# Functional characterization of *SMARCA4* variants identified by targeted exome-sequencing of 131,668 cancer patients

Tharu M. Fernando [1], Robert Piskol [2], Russell Bainer [2], Ethan S. Sokol[3], Sally E. Trabucco [3], Qing Zhang[4], Huong Trinh[4], Sophia Maund[5], Marc Kschonsak[6], Subhra Chaudhuri[7], Zora Modrusan[7], Thomas Januario[1] & Robert L. Yauch [1✉]

Genomic studies performed in cancer patients and tumor-derived cell lines have identified a high frequency of alterations in components of the mammalian switch/sucrose non-fermentable (mSWI/SNF or BAF) chromatin remodeling complex, including its core catalytic subunit, SMARCA4. Cells exhibiting loss of SMARCA4 rely on its paralog, SMARCA2, making SMARCA2 an attractive therapeutic target. Here we report the genomic profiling of solid tumors from 131,668 cancer patients, identifying 9434 patients with one or more *SMARCA4* gene alterations. Homozygous *SMARCA4* mutations were highly prevalent in certain tumor types, notably non-small cell lung cancer (NSCLC), and associated with reduced survival. The large sample size revealed previously uncharacterized hotspot missense mutations within the SMARCA4 helicase domain. Functional characterization of these mutations demonstrated markedly reduced remodeling activity. Surprisingly, a few SMARCA4 missense variants partially or fully rescued paralog dependency, underscoring that careful selection criteria must be employed to identify patients with inactivating, homozygous *SMARCA4* missense mutations who may benefit from SMARCA2-targeted therapy.

[1] Discovery Oncology, Genentech, South San Francisco, CA 94080, USA. [2] Bioinformatics and Computational Biology, Genentech, South San Francisco, CA 94080, USA. [3] Cancer Genomics Research, Foundation Medicine, Cambridge, MA 02141, USA. [4] Product Development Personalized Healthcare Data Science, Genentech, South San Francisco, CA 94080, USA. [5] Oncology Biomarker Development, Genentech, South San Francisco, CA 94080, USA. [6] Structural Biology, Genentech, South San Francisco, CA 94080, USA. [7] Molecular Biology, Genentech, South San Francisco, CA 94080, USA. ✉email: bobyauch@gene.com

The mammalian switch/sucrose non-fermentable (mSWI/SNF or BAF) complex is an ATP-dependent chromatin remodeler that uses the energy from ATP hydrolysis to slide, evict, deposit or alter the composition of nucleosomes, regulating the access of chromatin to other DNA-binding factors and transcriptional machinery[1,2]. Thus, it plays critical roles in development, differentiation and other important cellular processes like DNA replication and repair[3]. The BAF multimeric complex is formed by the combinatorial assembly of two mutually exclusive ATP-dependent helicases, SMARCA2 (BRM) and SMARCA4 (BRG1), with multiple accessory subunits that facilitate DNA- and histone-binding, allowing for extensive complex diversity and tissue-specific functions[4].

Cancer genomic studies in primary human tumors and tumor-derived cell lines revealed more than 20% of human tumors have mutations in one or more BAF subunits, with certain subunits found mutated in unique tumor types[5–9]. Many of these mutations are loss-of-function, and a large body of work has demonstrated that these complexes are in fact bona fide tumor suppressors[10–13]. Alterations in the core catalytic subunit, SMARCA4, have been found in multiple tumor types[14–19]. Recent studies have demonstrated that SMARCA4 mutations in the ATP-binding pocket fail to evict Polycomb repressive complex (PRC)-1 from chromatin and result in the loss of enhancer accessibility[7,8].

Strategies to therapeutically target BAF-mutant cancers have focused on identifying novel vulnerabilities due to the altered chromatin state caused by these mutations. Indeed, a subset of SMARCA4-deficient tumors were found to be sensitive to EZH2 inhibition, the catalytic subunit of PRC-2, with SMARCA2 expression potentially serving as a biomarker of insensitivity[20]. Synthetic lethal screens have also identified paralog dependence as an alternate vulnerability[21–25]. As BAF complexes have gained many paralogs that play distinct functions during development, somatic alterations in one paralog will result in a complete dependence on the remaining functional paralog for survival. Consequently, SMARCA2 has become an appealing therapeutic target in tumors that have mutation-driven loss of SMARCA4, and multiple efforts are ongoing to develop small molecule inhibitors of SMARCA2 activity or degraders[26–28].

Genomic studies thus far have described SMARCA4 alterations with limited patient data and have failed to assess differences in zygosity and co-occurrence with alterations in other BAF subunits and oncogenic drivers. However, to fully translate any potential SMARCA2-directed therapy into the clinic, it is imperative to understand the full spectrum of SMARCA4 mutations and their functional consequence. Here we explore SMARCA4 alterations in 131,668 cancer patients and functionally profile their remodeling activity and ability to compensate for SMARCA2 loss.

## Results

**SMARCA4 alteration spectrum in 131,668 patients with solid tumors.** To better characterize SMARCA4 somatic alterations, we analyzed targeted exome data of solid tumors from 131,668 cancer patients[29] and found SMARCA4 altered in 9,434 patients. SMARCA4 mutations were present in a diverse set of cancer types at frequencies up to 16% (Fig. 1a). More than half the mutations were missense (Fig. 1b), consistent with the spectrum of mutations described from The Cancer Genome Atlas (TCGA) and other pan-cancer analyses[5–8]. Higher tumor mutation burden (TMB) was found in the SMARCA4 variant population in all tumor types (Supplementary Fig. 1a). Overall, 90% of patients had only one SMARCA4 mutation (Supplementary Fig. 1b), although those with >1 SMARCA4 alteration had significantly higher TMB (Supplementary Fig. 1a).

Some indications like NSCLC and cancer of unknown primary (CUP) have a high prevalence of homozygous SMARCA4 mutations with >40% representing truncating alterations suggesting clear loss-of-function (Fig. 1c). This finding was further validated in NSCLC-derived cell line models where a subset harbor SMARCA4 mutations at high (>75%) variation frequency (Supplementary Fig. 1c). This observation is likely due to high rates of SMARCA4 loss-of-heterozygosity (LOH) found in NSCLC (77%) and CUP (68%) patients, which frequently co-occur with KEAP1 or STK11 alterations (all three genes are found on the same LOH segment), accounting for the majority of homozygous SMARCA4 alterations. Interestingly, homozygous SMARCA4 mutations were mutually exclusive with alterations in other BAF members (ARID1A, ARID1B, ARID2, PBRM1, SMARCB1 and SMARCD1) in NSCLC and CUP (Fig. 1d).

**SMARCA4 mutations are mutually exclusive with oncogenic drivers in NSCLC.** Due to the high prevalence of homozygous SMARCA4 alterations in NSCLC (10–25%) and the potential relevance of this population for SMARCA2 inhibition[26–28], we chose to further explore the mutational spectrum of SMARCA4 in NSCLC. 70-90% of SMARCA4 alterations were homozygous in NSCLC subtypes including the most common subtype, lung adenocarcinoma, with 15-40% representing truncating alterations (Supplementary Fig. 1d–e). With the emergence of novel targeted therapies in NSCLC, we evaluated whether SMARCA4 mutations co-occur with alterations in other actionable driver genes. Surprisingly SMARCA4 mutations were mutually exclusive with the most prevalent, targeted oncogenes in NSCLC, including EGFR, ALK, MET, ROS1 and RET ($P = 1.2E−34$). EGFR alterations demonstrated the strongest mutual exclusivity with SMARCA4 mutations ($OR = 0.280$, $P = 8.44E−42$), confirming previous reports that also found a significant anti-correlation in mutations of either gene[30–32] (Fig. 2a, b).

**NSCLC patients with homozygous SMARCA4 alterations have worse outcomes.** To understand if SMARCA4 alterations were associated with differences in clinical prognosis, we performed a retrospective study of a deidentified database of advanced diagnosis NSCLC patients (stage 3B+) treated in the Flatiron Health network between January 2011 and June 2017 who underwent FoundationOne® or FoundationOne® CDx tumor sequencing as part of routine clinical care. Because targeted therapy has substantially improved outcomes for patients with advanced diagnosis NSCLC, we focused our analysis on NSCLC patients who did not have known or likely driver mutations in EGFR, ALK, ROS1 or BRAF, which nevertheless are mutually exclusive with SMARCA4 alterations. We found that NSCLC patients with homozygous, truncating SMARCA4 mutations had significantly reduced overall survival (OS) compared to the wildtype (WT) SMARCA4 cohort (HR 1.85, $P < 0.0001$) (Fig. 2c). Because NSCLC patients will likely receive some form of checkpoint immunotherapy (CIT) targeting PD-1/PD-L1 in the course of their treatment, we also explored the outcome of SMARCA4-mutant patients treated with CIT. NSCLC patients with homozygous truncating SMARCA4 mutations had significantly worse OS on CIT compared to WT patients (HR 1.62, $P = 0.01$) (Fig. 2d). Interestingly this was despite SMARCA4-altered NSCLC patients having significantly increased TMB (a predictive biomarker for CIT response[33]) relative to the SMARCA4 WT population (Supplementary Fig. 1a). Collectively these studies indicate that advanced NSCLC patients with homozygous SMARCA4 truncating mutations represent a population with a clear unmet need that likely will not benefit from the currently available targeted molecular therapy and CIT.

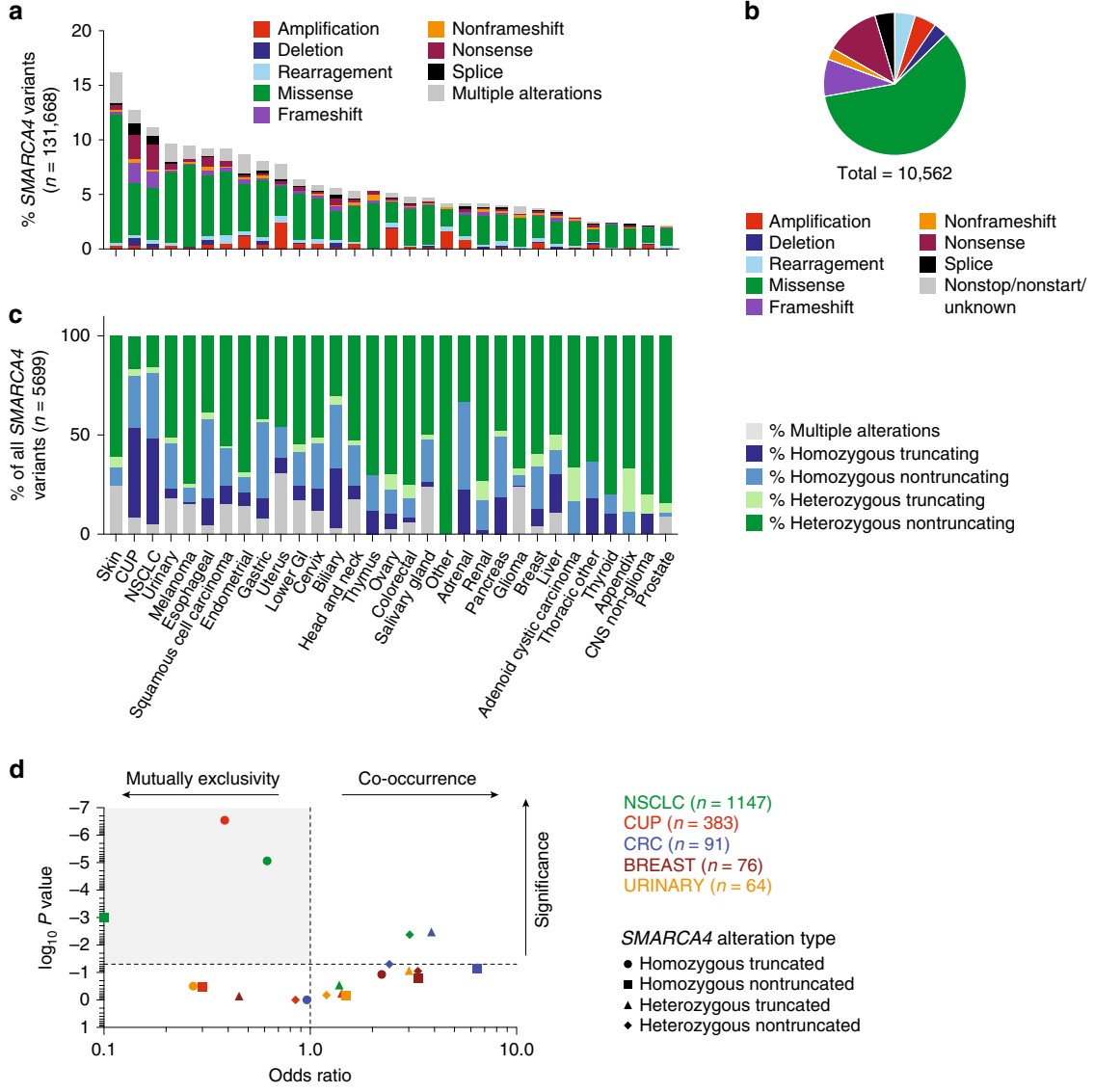

**Fig. 1 SMARCA4 mutational spectrum in the FoundationCORE® patient cohort. a** *SMARCA4* alteration frequency separated by disease ontology ($n = 131,668$ patients). **b** Distribution of *SMARCA4* mutation types ($n = 10,562$ variants). **c** Zygosity of *SMARCA4* truncating and nontruncating variants in patients where zygosity was determined ($n = 5699$ patients). **d** Mutual exclusivity of *SMARCA4* mutations with the aggregate of other BAF genes profiled in the FoundationOne® panel (*ARID1A/B, ARID2, PBRM1, SMARCB1, SMARCD1*).

**Identification of SMARCA4 hotspot mutations in the helicase domain**. While we highlight a subset of lung cancer patients with *SMARCA4* truncating mutations, almost 60% of *SMARCA4* alterations were missense mutations, and NSCLC patients with homozygous point mutations also trend towards reduced OS (HR 1.85, $P = 0.09$; Fig. 2c). Understanding the breadth of *SMARCA4* mutations and their functional consequence is crucial to identifying therapeutic strategies against these tumors. Previously only 927 *SMARCA4* variants have been identified[7,8], illustrating an incomplete picture of its mutational spectrum. By sequencing tumors from 131,668 patients, we have now identified 10,562 *SMARCA4* variants including 6,289 missense mutations. These data revealed previously described hotspots in the SNF2 domain[7,8] and additional hotspots in the C-terminal helicase domain (Fig. 3a). Hotspot missense mutations occurred within the ATP-binding cleft, DNA binding regions and brace helices (Fig. 3a, b). While some *SMARCA4* mutations within the ATP binding region have been previously characterized and deemed loss-of-function[7,8], it is unclear how the mutations that reside

outside of this region will affect protein activity. Interestingly, the most frequently mutated residues lie within highly conserved regions of *SMARCA4*, and certain residues within the ATP binding pocket (SMARCA4 A1186 and Arg finger R1192) and DNA binding contacts (SMARCA4 R1135 and R1157) are similarly mutated at equivalent sites in other SNF2 family helicases profiled in the FoundationOne® panel, like CHD4 and RAD54L (Supplementary Fig. 2a–c), signifying their potential functional importance. Many of these mutations are predicted to radically change the physiochemical properties of these residues by altering the charge (E821K; E882K; R1189Q; R1192C); adding bulky side chains (R1135W; R1243W); or modifying polarity (G1232S) (Fig. 3c).

**SMARCA4 missense mutants have reduced remodeling activity**. To better understand the consequence of *SMARCA4* missense mutations, we functionally characterized a panel of mutations found in the SNF2, C-terminal helicase domains and included a

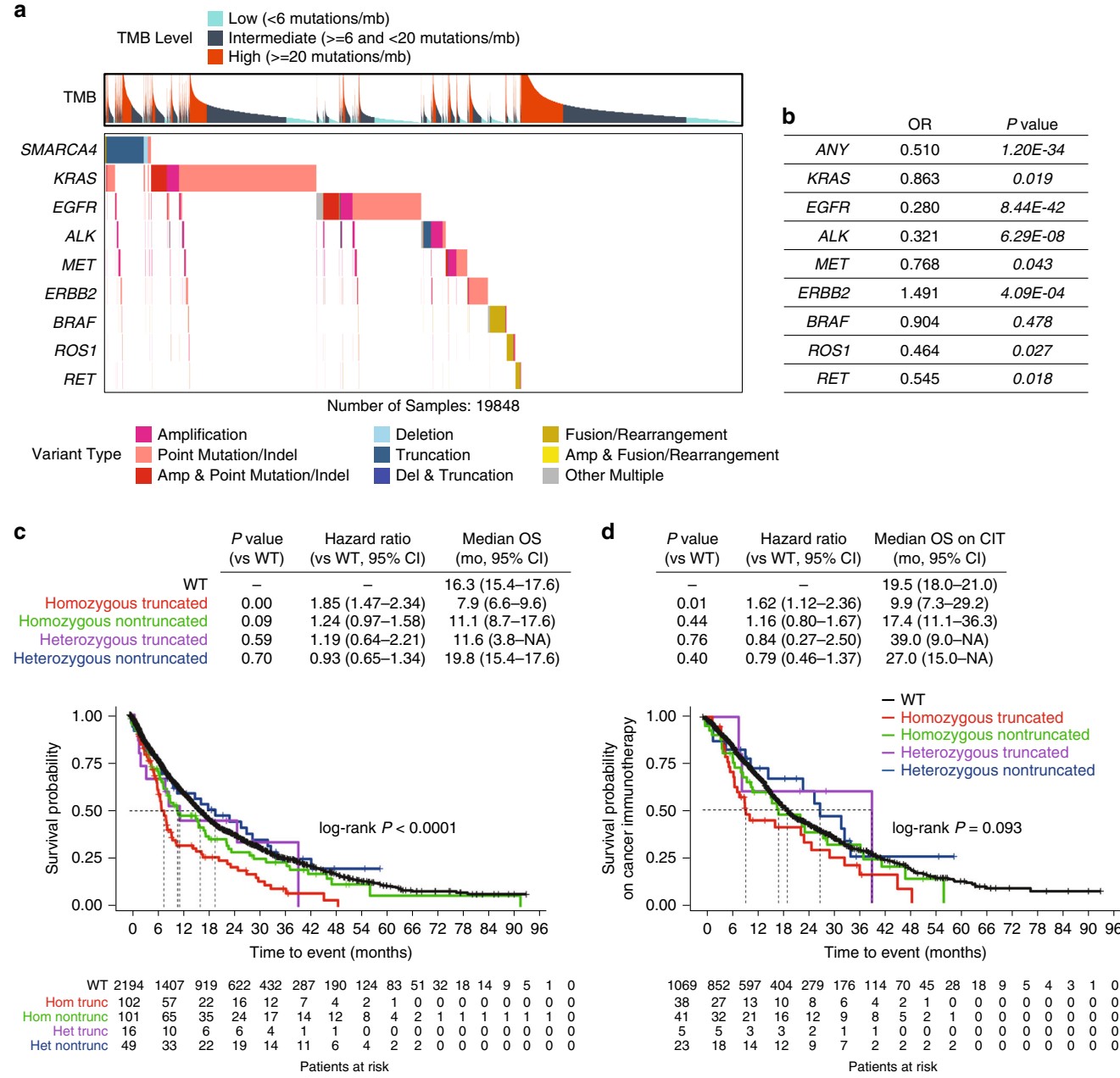

**Fig. 2 Homozygous SMARCA4 mutations in NSCLC. a** Co-occurrence of *SMARCA4* mutations with common oncogenic drivers in lung cancer patients excluding variants of unknown significance (*n* = 19,848 patients). **b** Summary table of odds ratio (OR) and *P* values using a two-tailed Fisher's exact test (not adjusted for multiple testing) for patients in **a**. **c** Overall survival of patients with *SMARCA4* mutations. **d** Overall survival of patients with *SMARCA4* mutations that have received cancer immunotherapy at any time during their treatment. The log-rank test was used to compare the overall survival of groups and resulting *P* values are unadjusted (**c–d**).

previously published ATPase-dead mutant (K785R)[34] as a control (Supplementary Fig. 2d). The biochemical compositions of immunopurified FLAG-tagged SMARCA4 mutant complexes were identical to WT and included BAF, polybromo-associated BAF (PBAF) and noncanonical (nc) BAF members (Supplementary Fig. 3a–c). SMARCA4 mutants were enriched in the insoluble chromatin fraction, suggesting that cellular localization was unaffected by the mutations (Supplementary Fig. 3d). Next, we assessed their ATP-dependent nucleosome remodeling function in vitro with fluorescence resonance energy transfer (FRET)- and gel shift-based nucleosome sliding assays. We found that only WT SMARCA4 was able to remodel nucleosomes in either assay (Fig. 4a; Supplementary Fig. 4), suggesting the mutants have

significantly reduced remodeling activity that is outside the detectable limits of both assays.

To uncover any residual activity the mutants may have in the context of chromatin, we tested their ability to alter chromatin accessibility by assaying transposase-accessible chromatin using sequencing (ATAC-seq) in SMARCA4-deficient NCI-H1944 cells transduced with SMARCA4 WT or mutants (Supplementary Fig. 5a). Reconstitution with SMARCA4 WT or mutants alone had no effect on cell growth (Supplementary Fig. 5b). However, the expression of WT led to a striking increase in chromatin accessibility with the AP-1 motif significantly enriched in these regions (Supplementary Fig. 5c), consistent with previously published studies reintroducing SMARCA4 or other BAF

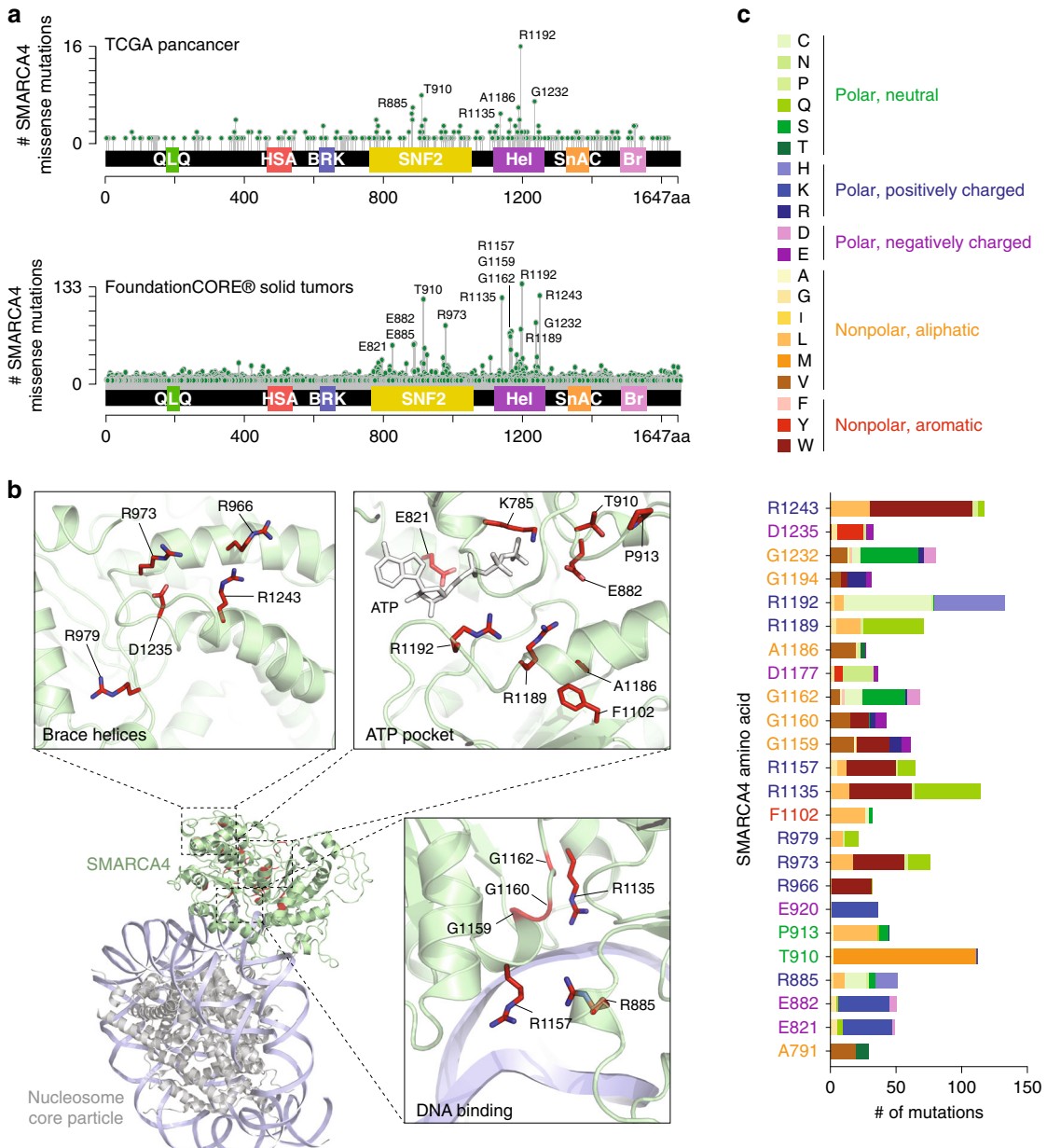

**Fig. 3 Hotspot missense mutations in the SMARCA4 ATPase helicase domain. a** Lollipop plots of SMARCA4 missense mutations in TCGA (top, $n = 379$ missense variants) and the FoundationCORE® (bottom, $n = 6289$ missense variants) cohorts. **b** Locations of frequently mutated residues on a SMARCA4 homology model derived from yeast Snf2-nucleosome complexes (PDB: 5X0X[48] and PDB:5Z3U[49]). **c** Amino acid changes of frequently mutated SMARCA4 residues.

subunits[35–39]. The increase in accessibility was associated with SMARCA4 occupancy and localized to intronic and intergenic regions (Supplementary Fig. 5d, e). SMARCA4 WT induced the expression of approximately 1000 genes, and Binding and Expression Target Analysis (BETA) demonstrated that upregulated genes were enriched for sites that had gained accessibility (Supplementary Fig. 5f, g).

Chromatin accessibility changes after reconstitution with SMARCA4 mutants were vastly different to changes seen after WT expression. While WT expression increased accessibility, mutant expression was deficient in this capacity and actually decreased accessibility at intronic and intergenic regions that were largely distinct from those opened by WT (Fig. 4b–d, Supplementary Fig. 5h). Sites with reduced accessibility in the mutant context disproportionally contained sequence motifs for HNF1B,

KLF5 and FOXA1 binding sites, as well as the AP-1 motif enriched in sites opened by WT (Supplementary Fig. 6a). Mutants A1186T and R973L had the lowest number of significantly closed ATAC sites with A1186T, even opening a few sites in contrast to the behavior of the other mutants (Fig. 4b). The observed decrease in accessibility with SMARCA4 mutants is consistent with a potential dominant negative function that has been previously described in the context of modeling SMARCA4 heterozygous mutant expression in embryonic stem cells, which do not express SMARCA2[7,8]. Because a large fraction of sites bound after SMARCA4 re-expression overlap with SMARCA2 binding sites (Supplementary Fig. 6b), we hypothesized that SMARCA4 mutants can partially interfere with the activity of its paralog. Indeed, we found that the sites closed by the mutants (cluster 1) are highly accessible in control (LACZ) cells and

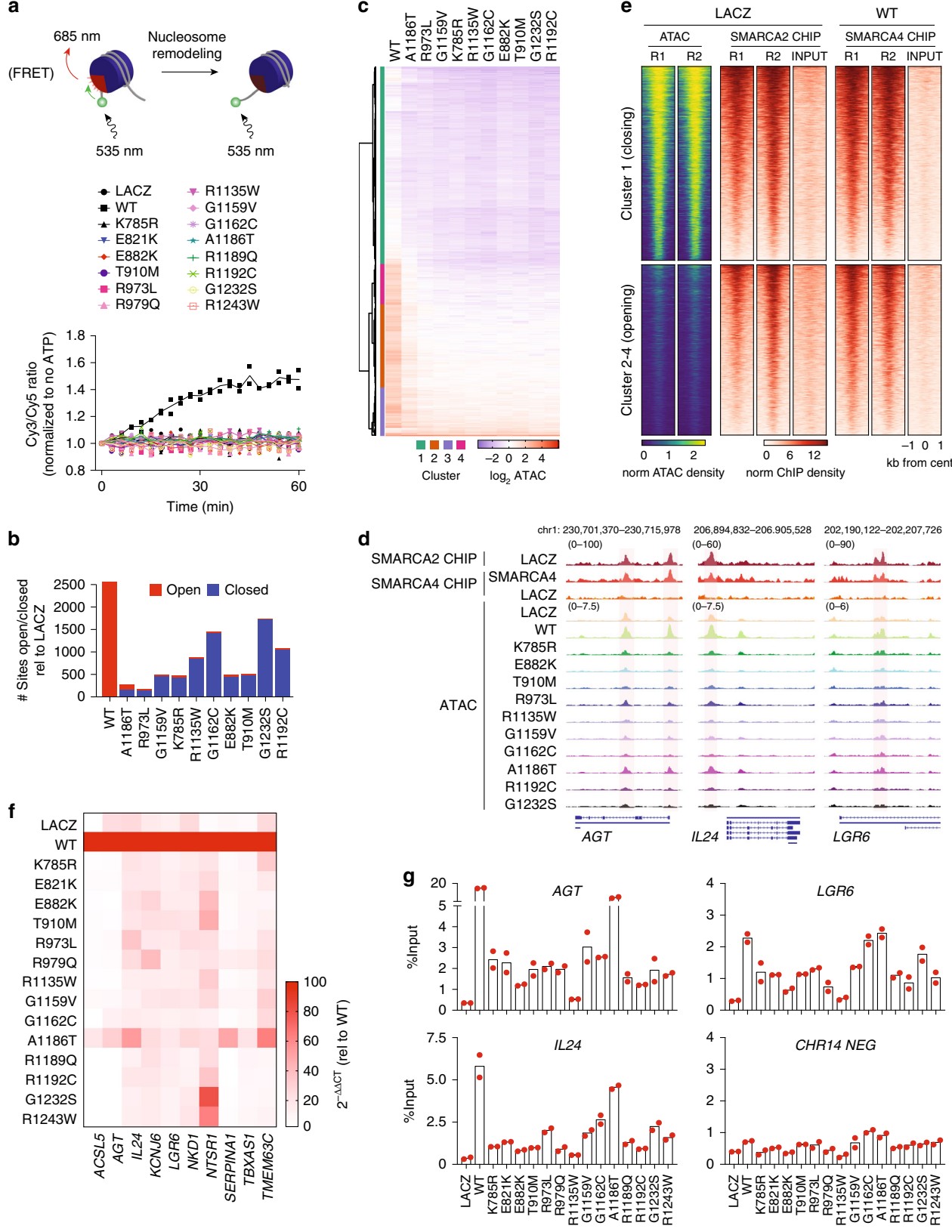

exhibited enrichment of SMARCA2 binding (Fig. 4e). Accessibility of these regions was reduced after *SMARCA2* knockdown in control cells, indicating these regions are maintained open by SMARCA2 (Supplementary Fig. 6c). In contrast, the regions opened by WT had low accessibility in the control cells, allowing for a gain in accessibility upon SMARCA4 binding (Fig. 4e). Reconstitution with the control, ATPase-dead K785R mutant resulted in more genes downregulated than upregulated, and BETA analysis demonstrated that downregulated genes were enriched for sites that had lost accessibility (Supplementary

**Fig. 4 SMARCA4 missense mutants are deficient in opening chromatin and inducing gene expression. a** FRET nucleosome remodeling assays were performed with immunopurified SMARCA4 WT and mutants from 293T cells transduced with SMARCA4 WT or mutants. Cy3/Cy5 ratios are represented in a 60 min kinetic assay, each construct is normalized to its no ATP control ($n = 2$ biologically independent samples, lines represent the mean). **b** Significantly open and closed sites as measured by ATAC-seq in NCI-H1944 cells expressing SMARCA4 WT or mutants relative to the LACZ control ($n = 2$ per construct). Significance was tested with a moderated t-statistic (two-sided) and P values were adjusted for multiple testing with the Benjamini–Hochberg procedure. **c** Heatmap of ATAC-seq changes relative to LACZ control ($\log_2$ fold-change) in the union of sites opened and closed from **b** ($n = 2$ per construct). **d** Representative IGV track of SMARCA2/SMARCA4 ChIP-seq and ATAC-seq changes in cells from **b** (overlay of 2 replicates per construct). **e** Heatmap of chromatin accessibility and SMARCA2 and SMARCA4 occupancy at sites from **c** in NCI-H1944 cells transduced with the LACZ control or SMARCA4 WT ($n = 2$ per construct). Data are shown as normalized peak counts per million genomic DNA fragments in a 2 kb window around the peak center. Rows are rank ordered by ATAC-seq peaks. R, replicate. **f** Heatmap of qRT-PCR analysis of a subset of SMARCA4 WT-induced genes in NCI-H1944 cells transduced with SMARCA4 WT or mutants (mean of $n = 3$ biologically independent samples). **g** SMARCA4 qChIP at target genes and a negative control region on chr14 in cells from **f** (each dot represents a technical replicate, $n = 2$; representative of 3 independent experiments). Source data are provided as a Source Data file.

Fig. 6d, e). These results are consistent with a model in which mutant SMARCA4 impairs the ability of endogenous SMARCA2 to maintain chromatin accessibility and expression of its targets.

The deficiency of SMARCA4 mutants to open chromatin would suggest that they are additionally defective in their ability to regulate the transcriptional changes observed upon SMARCA4 WT expression. We tested a panel of genes that exhibited increased accessibility and transcriptional changes upon SMARCA4 WT expression by qRT-PCR and observed that SMARCA4 mutants lacked the capacity to upregulate these transcripts to the same extent as WT (Fig. 4f). Interestingly SMARCA4 A1186T was the only mutant to modestly upregulate any of these genes and, at best, only up to 60%. This pattern was replicated upon testing a separate panel of genes that were upregulated by WT re-expression in another SMARCA4-deficient cell line, NCI-H1299 (Supplementary Fig. 6f–i). To determine whether the lack of accessibility and transcriptional regulatory activity was due to defects in chromatin binding, we performed qChIP on a few previously determined SMARCA4-bound sites and found that while nearly all mutants had enrichment above the LACZ control, they could not bind as well as WT, with the exception of the A1186T mutant (Fig. 4g). This observed decrease in binding perhaps captures defects in ATP hydrolysis or DNA-stimulated ATP hydrolysis, which can alter SMARCA4 chromatin dynamics. In line with this finding, the R1135W mutation lies within a DNA binding region of SMARCA4 and, as expected, exhibited a marked decrease in occupancy.

**SMARCA4 missense mutants have differing capacities to rescue SMARCA2 loss.** The ability of SMARCA2 to compensate for the loss of SMARCA4 has made SMARCA2 an attractive therapeutic target for SMARCA4-mutant tumor types, motivating multiple groups to generate SMARCA2 small molecule inhibitors or degraders[26–28]. Although cells harboring SMARCA4 homozygous truncating mutations are sensitive to SMARCA2 loss (Supplementary Fig. 7a, confirming previously published studies[21,23–25]), it is unclear if SMARCA4 missense mutants can compensate for SMARCA2 loss, which will have important implications in the future clinical development of SMARCA2-targeting agents. To this end, we knocked down SMARCA2 in SMARCA4-deficient NCI-H1944 and A549 cells and observed a significant decrease in growth, which was completely rescued with reintroduction of WT SMARCA4 (Fig. 5a, b, Supplementary Fig. 7b, c). The majority of SMARCA4 mutants tested were unable to rescue SMARCA2 knockdown, confirming that these mutants (K785R, E882K, T910M, R1135W, G1162C, R1192C, G1232S) are indeed loss-of-function (LOF) (Fig. 5a, Supplementary Fig. 7b). Surprisingly, a few SMARCA4 mutants either fully (A1186T) or partially rescued (R973L; G1159V) the growth defect observed after SMARCA2 depletion, despite having no

remodeling activity in vitro and negligible chromatin remodeling activity compared to WT in SMARCA2-proficient cells (Fig. 4a–d). We validated the rescue effects seen with SMARCA4 A1186T and R973L with CRISPR/Cas9 knockout of SMARCA2, and immunofluorescence demonstrated the residual cells that grew after SMARCA2 knockout were in fact SMARCA2 negative (Supplementary Fig. 7d, e). Taken together, these results suggest that the rescue in viability is conferred by hypomorphic activity of these two mutants and not due to incomplete SMARCA2 knockdown.

To understand how changes in accessibility might reflect the growth phenotype, we performed ATAC-seq with SMARCA4 WT and mutants after SMARCA2 knockdown in NCI-H1944 cells. We observed a marked decrease in chromatin accessibility after SMARCA2 depletion, which was completely rescued with SMARCA4 WT (Fig. 5c). Complementary ChIP-seq studies with a doxycycline (DOX)-inducible SMARCA2-targeting hairpin demonstrated SMARCA4 occupancy at these sites upon DOX treatment, suggesting that accessibility is maintained by direct binding of SMARCA4 (Fig. 5d). Notably, the A1186T and R973L mutant exhibited a marked ability to overcome the accessibility loss observed under the selective pressure of SMARCA2 depletion (Fig. 5c). The rescue in chromatin accessibility after SMARCA2 knockdown was well correlated with the rescue of the growth phenotype: LOF mutants, which had the strongest growth defect after shSMARCA2, also produced the largest decrease in accessibility (Fig. 5e). Surprisingly the decrease in accessibility observed after SMARCA2 knockdown was strongest in the LACZ control relative to the LOF mutant lines in both total ATAC read density and the number of sites lost (Supplementary Fig. 7f, Fig. 5e). These results suggest that the LOF mutants partially rescued the accessibility of a subset of SMARCA2-regulated sites. These results are consistent with previously described activity-independent sites maintained by SMARCA4 mutants, K785R and T910M, in ovarian cancer cell lines[40] and suggest that these sites are dispensable for cell viability. The SMARCA2 program not rescued by LOF mutants likely mediates the growth defect observed after SMARCA2 loss. In addition to the loss in accessibility, the K785R mutant failed to rescue the majority of the gene expression program lost after SMARCA2 depletion (Fig. 5f). These activity-dependent genes could further serve as biomarkers of potent SMARCA2 depletion as they include previously described SMARCA2 targets like KRT80[27].

Having observed a differential ability of particular missense mutations to compensate for SMARCA2 loss, we turned to a panel of cell line models harboring endogenous mutations to rule out the possibility that these effects are an artifact of over-expression systems. We found 5 cell lines that harbored endogenous SMARCA4 mutations (G1162C, A1186T and G1232S/D), 3 of which had homozygous mutations. CW-2 and

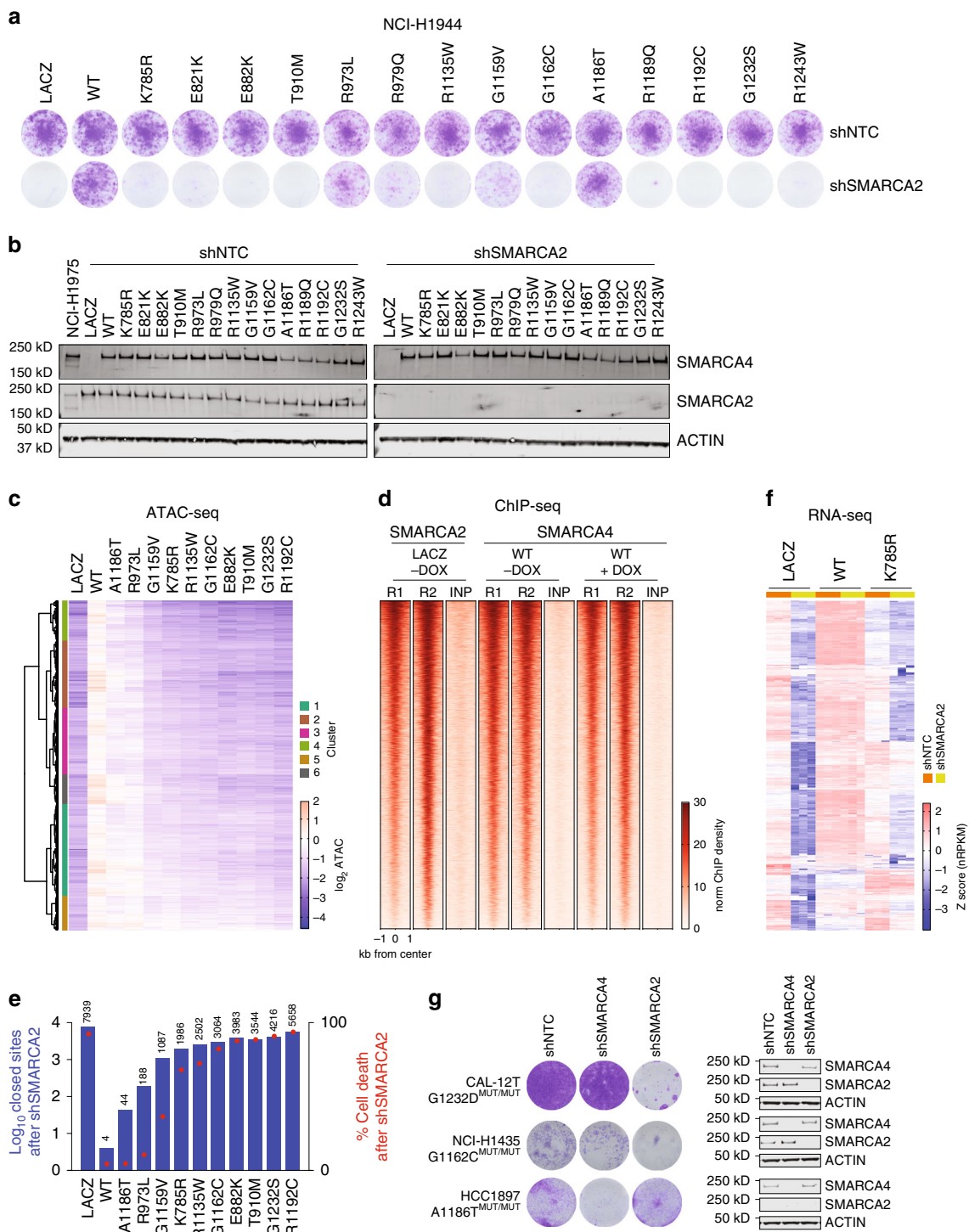

CAL 54 cells, which both had heterozygous mutations (G1232D, G1232S, respectively), were unaffected by *SMARCA2* knockdown (Supplementary Fig. 7g). In contrast, CAL-12T (G1232D) and NCI-H1435 (G1162C) cells expressed homozygous mutations and had severe growth defects when *SMARCA2* was depleted (Fig. 5g). Interestingly HCC1897 cells, which express SMARCA4 A1186T, were not dependent on *SMARCA2* but rather exhibited growth defects upon *SMARCA4* knockdown. Upon closer examination, we observed that HCC1987 cells did not express SMARCA2, a characteristic of sarcomatoid and/or thoracic sarcomas[19,41]. Taken in combination with the studies above, these data suggest that the endogenous SMARCA4 A1186T

mutant retains activity sufficient to confer viability in the absence of SMARCA2.

In light of the ongoing development of SMARCA2 inhibitors/degraders[26–28], our comprehensive exploration of the *SMARCA4* mutation landscape has provided some key insights for future patient selection strategies. The synthetic lethality conferred upon SMARCA2 depletion/inhibition requires the complete functional inactivation of SMARCA4. This finding calls for a careful interpretation of *SMARCA4* mutations by considering both the underlying genetics (i.e., zygosity), as well as the functional ability of individual mutations to compensate for paralog loss. By leveraging the largest cancer patient cohort described to-date, the

**Fig. 5 Differential effects of SMARCA4 mutants to rescue cell growth and chromatin accessibility loss after SMARCA2 knockdown. a** Long term clonogenic growth of NCI-H1944 cells transduced with SMARCA4 WT or mutants after *SMARCA2* knockdown. Representative of at least 3 replicates. **b** Immunoblot of cells from **a** (representative of at least 3 replicates). **c** Heatmap of ATAC-seq changes at sites after *SMARCA2* knockdown in cells from **a** ($n = 2$ per construct). Values represent $\log_2$ fold-change relative to LACZ control after *SMARCA2* knockdown. **d** Heatmap of SMARCA2 and SMARCA4 occupancy at regions with lower accessibility after *SMARCA2* knockdown (sites from **c**) ($n = 2$ per construct). SMARCA2 ChIP-seq was performed in NCI-H1944 cells expressing LACZ. SMARCA4 ChIP-seq was performed in NCI-H1944 cells expressing SMARCA4 WT and doxycycline (DOX)-inducible expression of *SMARCA2*-targeting shRNA. Data are shown as normalized peak counts per million genomic DNA fragments in a 2 kb window around the peak center. Rows are rank ordered by SMARCA2 enrichment. R, replicate; INP, input. **e** Number of sites closed (left axis, blue bar, $n = 2$ per construct) and mean percent cell death (right axis, red dot, mean of 3 replicates) after *SMARCA2* knockdown in cells from **a**. **f** Heatmap of genes downregulated after *SMARCA2* knockdown in NCI-H1944 cells transduced with LACZ, WT or K785R mutant ($n = 3$ per construct). Data are shown as mean-centered normalized reads per kb of transcript per million mapped reads (nRPKM). **g** Long term clonogenic growth of CAL-12T, NCI-H1435 and HCC1897, which all harbor homozygous *SMARCA4* missense mutations, after knockdown of *SMARCA2* or *SMARCA4* (left). Immunoblot confirming SMARCA2/SMARCA4 protein depletion (right). Data are representative of at least 2 replicates. Source data are provided as a Source Data file.

---

data provide clarity relative to previous reports on the frequency and characteristics of patients with biallelic, clear loss-of-function (truncating) mutations in *SMARCA4* that are potential candidates for future SMARCA2-targeting therapies. Furthermore, the importance of functional assessment is best highlighted by the identification of select, homozygous *SMARCA4* hotspot mutations that are largely deficient in chromatin remodeling activity but can confer hypomorphic function capable of maintaining cell viability under the selective pressure of SMARCA2 loss. Our data demonstrate the need to functionally assess variants of unknown significance more broadly in the future. Finally, one limitation of this study is our ability to address the potential for concurrent loss of *SMARCA2* in *SMARCA4*-mutant cancers. The previously described association of *SMARCA2* loss with rare BAF-deficient sarcomas[19,42] and/or lung sarcomatoid carcinomas[41] (the latter of which represented <1% of the lung cancers profiled, Supplementary Fig. 1d) would suggest it represents a minor percentage of SMARCA4-mutant cases[43,44], but nevertheless testing for SMARCA2 expression should be considered for future SMARCA2-targeted therapies.

## Methods

**Tumor samples and sequencing.** Samples were processed in the protocol developed for solid tumors as previously described[29]. Samples were submitted to a CLIA-certified, New York State-accredited, and CAP-accredited laboratory (Foundation Medicine, Inc., Cambridge, MA) for hybrid capture-based next-generation sequencing (NGS)-based genomic profiling. The pathologic diagnosis of each case was confirmed by review of hematoxylin and eosin stained slides, and all samples that advanced to nucleic acid extraction contained a minimum of 20% tumor cells. The samples used in this study were not specifically selected and represent an 'all comers' patient population to Foundation Medicine genomic profiling. For solid tumors, DNA was extracted from formalin fixed paraffin embedded 10 μm sections. Adaptor-ligated DNA underwent hybrid capture for all coding exons of 287 or 395 cancer-related genes plus select introns from 19 or 31 genes frequently rearranged in cancer. Captured libraries were sequenced to a median exon coverage depth of >500x (DNA) using Illumina sequencing, and resultant sequences were analyzed for base substitutions, small insertions and deletions (indels), copy number alterations (focal amplifications and homozygous deletions) and gene fusions/rearrangements, as previously described[29]. Frequent germline variants from the 1000 Genomes Project (dbSNP142) were removed. Zygosity of mutations was determined with the experimental somatic-germline-zygosity (SGZ) computational method, as previously described[45]. *SMARCA4* truncating alterations included frameshift indels, nonsense, or splice mutation types; *SMARCA4* nontruncating alterations included missense and nonframeshift indels. Tumor mutational load was calculated as the number of somatic base substitution or indel alterations per Mb of the coding region target territory of the test (currently 1.1 Mb). The data represent samples collected through Dec 2017 of the FoundationCORE® database ($n = 131,668$ total samples). *SMARCA4* variants identified and total number of tumor types profiled are found in Supplementary Data 1 and 2, respectively. Approval for this study was obtained from the Western Institutional Review Board (protocol number 20152817). Patients consented for the use of their data for analysis but not for raw data release.

**Mutual exclusivity analyses.** Mutual exclusivity analyses of *SMARCA4* with other BAF members (*ARID1A*, *ARID1B*, *ARID2*, *PBRM1*, *SMARCB1*, *SMARCD1*) or with actionable oncogenes in NSCLC (*EGFR*, *KRAS*, *BRAF*, *ALK*, *ROS1*, *RET*, *ERBB2*,

*MET*) were performed on samples with zygosity determined (as described above) and excluded variants of unknown significance. *SMARCA4* truncating alterations included frameshift indels, nonsense, or splice mutation types; *SMARCA4* non-truncating alterations included missense and nonframeshift indels. Odds ratio and *P* values of co-occurrence was calculated using the Fisher's exact test. Odds ratios >1 indicate co-occurrence and <1 indicate mutual exclusivity.

**Kaplan–Meier survival analyses.** Kaplan-Meier survival analyses were performed on a sample of patients with advanced diagnosis NSCLC extracted from a dei-dentified database previously described[46]. Patients treated in the Flatiron Health network (>265 oncology practices across the U.S.) between Jan 2011 and April 2019 who underwent comprehensive genomic profiling by Foundation Medicine as part of routine care were eligible. The advanced diagnosis NSCLC patient cohort was defined by patients with an advanced diagnosis NSCLC stage 3B+ no earlier than January 2011; who encountered their first line of therapy within 90 days of advanced diagnosis; and received commercial genomic profiling no earlier than 90 days before advanced diagnosis. Patients with alterations in *EGFR*, *ALK*, *ROS1* and *BRAF* alterations of known or likely function were excluded from the advanced diagnosis NSCLC patient cohort to eliminate receiving targeted therapy as a confounding factor. Patients were then stratified based on the zygosity and type of *SMARCA4* alteration. Survival analysis on cancer immunotherapy was performed by selecting patients from the advanced diagnosis NSCLC patient cohort that received Nivolumab, Pembrolizumab, Atezolizumab or Durvalumab at any time during the course of their treatment after advanced diagnosis. The log-rank test was used to compare the overall survival of groups and resulting *P* values are unadjusted. Institutional Review Board approval of the study protocol was obtained prior to study conduct. Informed consent was waived as this was a non-interventional study and the anonymized data in the Flatiron-Foundation Medicine Clinico-Genomic database are protected against breach of confidentiality.

**SMARCA4 variant frequency in human tumor-derived cell lines.** *SMARCA4* variant frequency was determined from exome-seq done on cell lines from the Genentech cell bank (gCell). Cell lines with *SMARCA4* splice region variants, mutations in known SNP variants and those with <2% *SMARCA4* variation frequency were excluded. A total of 98 cell lines were used for this analysis (Supplementary Data 3).

**SMARCA4 homology model.** The homology model was generated using the SMARCA4 sequence (isoform 2, UniProt: P51532-2) by submitting it to the SWISS-MODEL automated structure homology-modeling server[47]. The model was built based on the Snf2 of the yeast Snf2-nucleosome cryo-EM structure (PDB: 5X0X[48]) with a sequence identity of 58.5%. The SMARCA4 homology model was then aligned to the yeast Snf2 structure-nucleosome complex bound to ADP-BeFx (PDB:5Z3U[49]) and an ATP molecule was placed at the position of the ADP-BeFx. The figures were generated using PyMOL (Version 2.0 Schrödinger, LLC).

**SMARCA4 helicase domain conservation scores.** The conservation score at each residue of the SMARCA4 helicase domain was generated by performing multiple sequence alignment on 233 SMARCA4 ortholog protein sequences (OMA Group 572177). The alignment of SMARCA4 residues 753-1301 was then used to score sequence conservation based on Jensen-Shannon divergence using a three-residue averaging window[50]. Mutation counts were determined by counting the absolute number of mutations that occurred at each residue of SMARCA4 in the Foundation Medicine tumor samples described above.

**Alignment of P-loop containing ATPase.** Due to the large number of tumor samples in the Foundation Medicine cohort, we chose to identify other mutations found in SNF2 family ATP-dependent helicases profiled in the FoundationOne® gene panel. These included *ATRX*, *CHD2*, *CHD4* and *RAD54L*. The P-loops of the

ATPase domains were used for multiple sequence alignment using ClustalOmega (version 1.2.2)[51]. Alignments were manually matched with residues that had at least 10 missense variants. Lollipop plots were generated for each helicase using the cBioPortal mutation mapper[52,53].

**Cell lines.** All cell lines were grown in RPMI 1640 supplemented with 10% fetal bovine serum (FBS), 2 mM L-Glutamine and 100 U/mL penicillin-streptomycin (Gibco) unless otherwise stated. A549, NCI-H838, NCI-H1299, NCI-H1435, NCI-H1793, NCI-H1944 and NCI-H1975 cells were obtained from ATCC. CW-2 cells were obtained from the Riken Institute. HCC1897 cells were obtained from University of Texas Southwestern Medical Center. CAL 54 and CAL-12T cells were obtained from DSMZ and grown in DMEM supplemented with 10% FBS, 2 mM L-Glutamine, and 100 U/mL penicillin-streptomycin (Gibco). Lenti-X 293T cells were obtained from Takara Bio and grown in DMEM supplemented with 10% FBS, 2 mM L-Glutamine, 1X MEM non-essential amino acids (Gibco), 1 mM sodium pyruvate (Gibco) and 100 U/mL penicillin–streptomycin (Gibco). All cell lines were authenticated using SNP genotyping and STR profiling by the Genentech internal cell line repository, gCell, and used for experiments within 15 passages.

**Lentiviral constructs and infection.** The pLenti6.3 vector system was used for all overexpression experiments. *SMARCA4* NM_001128847.1 (NP_001122319.1) was used to generate FLAG-tagged *SMARCA4* WT and mutants (K785E, E821K, E882K, T910M, R973L, R979Q, R1135W, G1159V, G1162C, A1186T, R1189Q, R1192C, G1232S and R1243) in pLenti6.3 (GenScript Biotech). pLent6.3 containing *LACZ* was used as a control. FLAG-constructs were used to reconstitute SMARCA4-deficient A549 (*SMARCA4* frameshift variant), NCI-H1944 (*SMARCA4* gene deletion) and NCI-H1299 (*SMARCA4* frameshift variant). Constitutive knockdown was achieved using shRNAs directed against nontargeting control (5'-AACCACGTGAGGCATCCAGGC-3'), *SMARCA2* (5'-TCGTCGAG-CAATCATTTGGTT-3') and *SMARCA4* (5'- TAGCATTGAGGCTGTCTCCA -3') in a modified pLKO lentiviral vector, which uses the miR-3G hairpin expression[54]. The packaging and envelope vectors, Δ8.91 and VSV.G were co-transfected with pLenti6.3- or pLKO-based constructs into Lenti-X 293T cells using Fugene 6 (Promega). Media containing lentiviral particles was collected 48 or 72 h post transfection, filtered through 0.45 μm filters and either concentrated using Lenti-X concentrator (Takara Bio) or used directly to transduce cell lines. Cells were transduced with pLenti6.3-*SMARCA4* WT and mutant constructs and selected with 8 μg/mL blasticidin (Gibco) for at least 5 d before use in other experiments or infected with shRNAs. Cells were infected with shRNAs in the pLKO backbone and then selected with 3 μg/mL puromycin (Gibco) for at least 2 d before being plated for proliferation assays. To generate doxycycline-inducible knockdown of *SMARCA2* for ChIP experiments, the *SMARCA2* shRNA from above was cloned into the pINDUCER10, and cells were transduced as above and selected with 1.5 μg/mL puromycin (Gibco). Cells stably expressing inducible constructs were subcloned to select for clones that had minimal leakiness, these clones were then transduced with pLenti6.3-*LACZ* or *SMARCA4* WT and selected with 8 μg/mL blasticidin (Gibco) for at least 5 d before being scaled up for ChIP experiments.

**Proliferation and colony forming assays.** SMARCA4 WT- and mutant-expressing cells (A549: 500 C/well; NCI-H1944: 1000 C/well) were plated in black clear bottom 96-well plates (BD Falcon) and confluence was measured over time in the Incucyte (Essen Biosciences). Cells (A549: 500 C/well; NCI-H1944: 5000 C/well; NCI-H838: 3000 C/well; NCI-H1435: 15,000 C/well; NCI-H1793: 3000 C/well; CW-2: 15,000 C/well; CAL-12T: 5000 C/well; CAL 54: 5000 C/well; HCC1897: 15,000 C/well) were plated in 12-well plates and assayed for long term growth for 10-14 d. Cells were visualized by staining with 0.5% crystal violet solution containing 20% methanol for 20 min at room temperature.

**Whole cell lysate.** Cells were washed once in PBS, scraped and lysed in safe-lock Eppendorf tubes with modified RIPA buffer (10 mM Tris pH 7.4, 150 mM NaCl, 2 mM EDTA, 1% Igepal CA-630, 0.1% SDS) supplemented with Halt EDTA-free protease and phosphatase inhibitor cocktail (Pierce). A 3.2 mm stainless steel homogenization bead (NextAdvance) was added to the lysate and then homogenized for 3 min at speed 10 using the Bullet Blender (NextAdvance). Protein was cleared by centrifugation 20,000×g for 15 min at 4 °C and quantified using the BCA assay (Pierce).

**Subcellular fractionation.** $1 \times 10^7$ cells were washed once with PBS, scraped and resuspended in buffer A (10 mM HEPES pH 7.9, 10 mM KCl, 1.5 mM MgCl$_2$, 0.34 M sucrose, 10% glycerol, 1 mM DTT and Halt EDTA-free protease and phosphatase inhibitor cocktail (Pierce)). 0.1% Triton X-100 was added from a 10% stock solution to the lysate, which was then incubated on ice for 5 min and spun at 1300×g for 4 min at 4 °C. Cytosolic fraction was transferred to new tube. Nuclei were washed in buffer A (no Triton X-100) and spun at 1300×g for 4 min at 4 °C. Nuclei was resuspended in buffer B (3 mM EDTA, 0.2 mM EGTA, 1 mM DTT and Halt EDTA-free protease and phosphatase inhibitor cocktail), incubated on ice for 30 min and spun at 1700×g for 4 min at 4 °C. Soluble nuclear fraction was transferred to new tube, and insoluble chromatin pellet was washed in buffer B and then

spun at 1,700xg for 4 min at 4 °C. Insoluble chromatin pellet was resuspended in lysis buffer (0.5 M NaCl, 1% Triton X-100, 0.1% SDS and Halt EDTA-free protease and phosphatase inhibitor cocktail) and homogenized in a Bullet Blender (NextAdvance) for 3 min at speed 10 with a 3.2 mm stainless steel homogenization bead (NextAdvance).

**Purification of FLAG complexes.** Lenti-X 293T cells expressing FLAG-tagged SMARCA4 constructs were expanded to $3 \times 150$ mm dishes and allowed to come to confluency. Cells were scraped, washed with cold PBS and lysed with Triton lysis buffer (50 mM Tris-HCl pH 7.4, 150 mM NaCl, 2 mM MgCl$_2$, 1% Triton X-100, Halt EDTA-free protease and phosphatase inhibitor cocktail (Pierce), and 10 U/mL universal nuclease (Pierce)). The lysate was rocked for 0.5–1 h at 4 °C and then spun at 20,000×g for 4 min at 4 °C. Cleared lysate was incubated with FLAG M2 affinity gel (Sigma) 2 h or overnight at 4 °C. The affinity gel was washed for 5 min rocking at 4 °C twice each with Triton lysis buffer, 300 mM NaCl wash buffer (Triton lysis buffer supplemented with 150 mM NaCl), 500 mM NaCl wash buffer followed by two quick TBS washes. FLAG complexes were eluted twice with elution buffer (20 mM Tris-HCl pH 7.4, 150 mM NaCl, 1 mM EDA, 10% Triton X-100, 0.1% Igepal CA-630, 1 mM DTT, Halt EDTA-free protease and phosphatase inhibitor cocktail and 0.15 mg/mL 3× FLAG peptide) for 0.5-1 h rocking at 4 °C. Eluates were concentrated using 10K MWCO protein concentrators (Amicon). Aliquots were flash frozen and stored at −80 °C.

**Silver stain.** An aliquot (~2–5%) of FLAG-purified complexes was prepared with NuPAGE LDS Sample Buffer and Sample Reducing Agent, heated for 5 min at 95 °C or 10 min at 70 °C and run on either NuPAGE 4–12% Bis-Tris or 3–8% Tris-Acetate protein gels (Invitrogen). Gels were rinsed with ultrapure water and incubated with fixative (40% ethanol, 10% acetic) for 20 min. Silver staining was performed using the SilverQuest Silver Staining Kit (Invitrogen), according to the manufacturer's protocol.

**Immunoblot.** Protein samples were prepared with NuPAGE LDS Sample Buffer (Invitrogen) and NuPAGE Sample Reducing Agent, heated for 5 min at 95 °C or 10 min at 70 °C and run on NuPAGE 3-8% Tris-Acetate protein gels (Invitrogen). Gels were transferred onto nitrocellulose membranes using the iBlot 2 Dry Blotting System (Invitrogen) at 20 V for 13 min Membranes were blocked with Starting-block (TBS) (ThermoFisher) for at least 30 min at room temperature before applying primary antibodies diluted in Startingblock and incubated overnight rocking at 4 °C. Membranes were washed with TBS supplemented with 0.1% Tween-20 (TBS-T) and IRDye 680RD- or IRDye 800CW-conjugated anti-rabbit IgG or anti-mouse IgG secondary antibodies (Licor) were applied. Membranes were washed again with TBS-T, TBS and visualized on the Licor Odyssey using Image Studio v3.1. The following primary antibodies were used at 1:1000: SMARCA4/BRG1 (Abcam ab110641); SMARCA2/BRM (Cell Signaling Technologies, 11966); SMARCC1/BAF155 (Cell Signaling Technologies, 11956 S); SMARCC2/BAF170 (Bethyl, A301-039A); BAF47/SNF5 (Cell Signaling Technologies, 91735); SMARCD1/BAF60A (Bethyl, A301-595A); SMARCE1/BAF57 (Bethyl, A300-810A); ACTL6A/BAF53A (Bethyl, A301-391A); ARID1A/BAF250A (Cell Signaling Technologies, 12354); ARID1B/BAF250B (Bethyl, A301-047A); SS18 (Cell Signaling Technologies, 21792); DPF2 (Abcam, ab134942); PBRM1/BAF180 (Millipore, ABE370); ARID2 (Santa Cruz, E-3); ARID2 (Bethyl, A302-230A); BRD7 (Cell Signaling Technologies, 15125); PHF10 (Abcam, ab154637); BRD9 (Abcam, ab137245); GLTSCR1 (Invitrogen, PA5-63267); GLTSCR1L (Novus, NBP1-86359); ACTIN (Cell Signaling Technologies, 3700), TUBULIN (Cell Signaling Technologies, 2148), HDAC1 (Cell Signaling Technologies, 34589), LAMIN A/C (Cell Signaling Technologies, 4777), FLAG (Sigma, M2). The following secondary antibodies were used at 1:1000: goat anti-mouse IgG conjugated to IRDye 680RD (Licor, 926-68070); goat anti-rabbit IgG conjugated to IRDye 800CW (Licor, 926-65010).

**Gel shift nucleosome remodeling assays.** Nucleosome reconstitution and gel shift remodeling assays were performed as previously described[55]. Sliding reactions were done with a 1:1 ratio of 20 nM Cy3-labeled center- and Cy5-edge-positioned nucleosomes in 20 mM HEPES pH 7.9, 40 mM KCl, 3 mM MgCl$_2$, 10% glycerol, 0.02% IGEPAL CA-630 with and without 2 mM ATP (Invitrogen). Reactions were started by the addition of 30 nM recombinant ACF (EpiCypher) or 4 μg FLAG-purified complexes (diluted in ACF remodeling assay buffer, EpiCypher) and occurred for 30 min at 30 °C. Reactions were stopped by adding salmon sperm (Invitrogen) to a concentration of 5 mg/mL. Samples were supplemented with Novex Hi-density TBE sample buffer (Invitrogen) and separated on 6% DNA retardation gels (Invitrogen) in 0.5X TBE. Nucleosome bands were visualized on a Typhoon Trio (GE Healthcare Life Sciences).

**FRET-based nucleosomes sliding.** Remodeling assays were performed with 20 nM EpiDyne FRET nucleosome remodeling substrate (EpiCypher) in 20 mM Tris pH 7.4, 50 mM KCl, 3 mM MgCl$_2$, 0.1 mg/mL BSA, 0.02% IGEPAL CA-630 with 4 μg FLAG-purified complexes (diluted in ACF remodeling assay buffer, EpiCypher) in 384-well white proxiplates (PerkinElmer). Reactions were initiated by adding 2 mM ATP (Invitrogen) and fluorescence intensity was immediately read on the

Tecan Infinite M-1000 with Tecan i-control software (common version 3.7.3.0), using excitation at 535 nm and reading emission at 579 (for Cy3) and 683 nm (for Cy5), every 3 min for 60 min.

**CRISPR-Cas9 knockout of SMARCA2**. Alt-R CRISPR-Cas9 crRNAs (XT version) for *SMARCA2* (5′-CTCCCAGTCCTACTACACCG-3′ and 5′-GTGA-CAGTTTCTCAGCGGG-3′) and negative control #1, Alt-R CRISPR-Cas9 tracrRNA, Alt-R *S. pyogenes* Hifi Cas9 Nuclease V3 and Alt-R Cas9 electroporation enhancer were purchased from IDT. Delivery of Cas9 ribonucleoproteins (RNP) complexes were performed according to IDT protocols using the Neon Transfection System (Invitrogen). Briefly, equimolar amounts of crRNA and tracrRNA were mixed to final duplex concentration of 44 μM in IDT duplex buffer, heated for 5 min at 95 °C and cooled slowly to room temperature. Cas9 RNPs were formed with 22 pmol of crRNA:tracrRNA duplexes and 18 pmol diluted Alt-R Cas9 enzyme and incubated for 20 min at room temperature. NCI-H1944 cells expressing SMARCA4 WT or mutants were trypsinized, counted and $1 \times 10^5$ cells/transfection were washed in PBS and resuspended in 9 μL Neon Resuspension Buffer R. For each transfection, cells were mixed with 1 μL RNP complex (1 μL of negative control #1 guide RNA; or 0.5 μL of both SMARCA2 guide RNAs) and 2 μL of electroporation enhancer (diluted to 10.8 μM). Cells were transfected in a 10 μL Neon tip for 2 pulses at 1400 V with a 20 ms pulse width and transferred to pre-warmed media in 6-well plates. 8 d post transfection, cells were plated in bulk for other assays.

**Immunofluorescence**. NCI-H1944 cells expressing SMARCA4 WT or mutants that had CRISPR-Cas9 knockout of *SMARCA2* or negative control were plated in black clear bottom 96-well plates (BD Falcon) at 1000 C/well. Cells were allowed to grow for 10 d and then fixed with 4% formaldehyde diluted in PBS. Cells were washed three times with PBS and blocked for 1 h in blocking buffer (10% FBS, 1% BSA, 0.1% Triton X-100, 0.01% sodium azide in PBS) at room temperature before applying primary antibodies diluted in blocking buffer (1:2000 anti-SMARCA2, Cell Signaling Technologies, 11966; 1:500 anti-SMARCA4, Santa Cruz, G-7) overnight at 4 °C. Cells were incubated with secondary antibodies at 1:1000 (Cell Signaling Technologies 4412, goat anti-rabbit IgG F(ab')₂ fragment conjugated with Alexa Fluor 488; and Cell Signaling Technologies 4410, goat anti-mouse IgG F (ab')₂ fragment conjugated with Alexa Fluor 647) for 1 h at room temperature in the dark. 0.5 μg/mL DAPI was added in the last 10 min of the secondary incubation. Cells were washed three times with PBS and left in PBS. Immunofluorescence was visualized using the Opera Phenix High Content Screening System (PerkinElmer).

**qRT-PCR**. RNA was isolated using the RNeasy Plus Mini kit (Qiagen) according to the manufacturer's instructions and quantified using the NanoDrop Spectrophotometer (ThermoFisher). Gene expression levels were determined with 50 ng of RNA per well, TaqMan gene expression assays (Applied Biosystems, found in Supplementary Data 4) and the TaqMan RNA-to-Ct 1-Step enzyme kit (Applied Biosystems). Analysis was performed using the QuantStudio 7 Flex Real-Time PCR system (Applied Biosystems).

**ChIP-PCR (qChIP)**. NCI-H1944 cells expressing SMARCA4 WT or mutants were grown to confluence in 150 mm dishes. Cells were fixed with 1% formaldehyde (Sigma) for 10 min, quenched with 0.125 M glycine for 10 min, washed with cold PBS three times and resuspended in shearing buffer (supplemented with Halt EDTA-free protease and phosphatase inhibitor cocktail) from the truCHIP Chromatin Shearing Kit (Covaris) and sonicated in a 1 mL milliTUBE with AFA fiber (Covaris) using the E220 focused-ultrasonicator (Covaris) for 20 min (with 5.0 acoustic duty factor; 140 peak incident power, 200 cycles/burst). 30 μg of sheared chromatin was incubated with 4 μL of SMARCA4 antibody (Abcam ab110641) preconjugated to 50 μL Protein A dynabeads (Invitrogen) in each immunoprecipitation. Immunoprecipitations occurred in 150 mM NaCl, 1% Triton X-100, 0.1% sodium deoxycholate, 0.1% SDS, 20 mM Tris-HCl pH 8.0, 1 mM EDTA, supplemented with Halt EDTA-free protease and phosphatase inhibitor cocktail at 4 °C overnight. Immunoprecipitations were washed using low salt (150 mM NaCl, 20 mM Tris-HCl pH 8.0, 0.1% SDS, 1% Triton X-100, 2 mM EDTA), high salt (500 mM NaCl, 20 mM Tris-HCl pH 8.0, 0.1% SDS, 1% Triton X-100, 2 mM EDTA), LiCl (250 mM LiCl, 10 mM Tris-HCl pH 8.0, 1% IGEPAL CA-630, 1 mM EDTA) and TE (10 mM Tris-HCl pH 8.0, 1 mM EDTA) wash buffers. Immunoprecipitated chromatin was eluted using elution buffer (1% SDS, 0.1 M sodium bicarbonate), reverse crosslinked, digested with 40 μg proteinase K at 65 °C overnight and purified using the Qiaquick PCR purification kit (Qiagen). qPCR was performed with purified DNA, 0.5 μM primers (Supplementary Data 4) and Fast SYBR Green master mix (Applied Biosystems) and analyzed using the QuantStudio 7 Flex Real-Time PCR system (Applied Biosystems).

**ChIP-seq**. NCI-H1944 cells ($20 \times 10^6$) stably transduced with doxycycline-inducible *shSMARCA2* and either LACZ or SMARCA4 WT were treated with vehicle or 0.5 μg/mL doxycycline (Clontech) for 4 d to obtain significant *SMARCA2* knockdown. Cells were fixed with 1% formaldehyde (Sigma) for 10 min, quenched with 0.125 M glycine for 10 min, washed with cold PBS three times and snap frozen. ChIP for SMARCA2 and SMARCA4 was performed by Active Motif

Epigenetic Services. Chromatin was isolated with the addition of lysis buffer followed by disruption with a Dounce homogenizer. Lysates were sonicated, and DNA was sheared to an average length of 300-500 bp. ChIPs were performed with 30 μg of precleared chromatin and 5 μl of anti-SMARCA2 (Abcam, ab15597) or 10 μL anti-SMARCA4 antibody (Abcam, ab110641). Complexes were washed, eluted from the beads with SDS buffer and subjected to RNase and Proteinase K treatment. Crosslinks were reversed overnight at 65 °C, and DNA was purified by phenol-chloroform extraction and ethanol precipitation. Illumina-compatible libraries were generated using an automated system (Apollo 342, Wafergen Biosytems/Takara) and sequenced on the Illumina NextSeq 500 (single-end 75 bp reads).

Sequencing reads were aligned to the human reference genome (NCBI Build 38) using GSNAP[56] version '2013-10-10', allowing a maximum of two mismatches per read sequence (parameters: '-M 2 -n 10 -B 2 -i 1 --pairmax-dna=1000 --terminal-threshold=1000 --gmap-mode=none --clip-overlap'). Mapped reads then were assessed for peaks relative to the input controls using Macs2 (version 2.1.0) callpeak function[57]. Peak-fold enrichment was calculated using Macs2, using a sliding window across the genome and assessing read counts relative to expected background. The Integrative Genomics Viewer (IGV) was used to visualize tracks.

**ATAC-seq**. NCI-H1944 cells (100,000) transduced with LACZ, SMARCA4 WT or mutant were pelleted and resuspended in 50 mL cold ATAC-resuspension buffer (ATAC-RSB) (10 mM Tris HCl pH 7.4, 10 mM NaCl, 3 mM MgCl₂) containing 0.1% Igepal CA-630, 0.1% Tween-20 and 0.01% digitonin (Promega). Cells were incubated on ice for 3 min and lysis buffer was washed out using 1 mL cold ATAC-RSB containing 0.1% Tween-20. Cells were inverted several times and nuclei was pelleted at 500×g for 10 min at 4 °C. Supernatant was discarded and nuclei were resuspended in 50 μl transposition mixture (2.5 μL Tn5 transposase, 25 μL 2× TD buffer, 16.5 μL PBS, 0.5 μL 1% digitonin, 0.5 μL 10% Tween-20, 5 μL H₂O) (Illumina). The transposase reaction was performed for 30 min at 37 °C in a thermomixer with 1000 rpm. DNA was purified using the MinElute purification kit (QIAGEN). Illumina-compatible libraries were generated as previously described[58,59] and sequenced on the Illumina HiSeq4000.

An average of 45 million paired-end reads (50 bp) per sample were obtained for each sample. GSNAP[56] (version 2013-10-10), allowing a maximum of two mismatches per read sequence (parameters: '-M 2 -n 10 -B 2 -i 1 --pairmax-dna=1000 --terminal-threshold=1000 --gmap-mode=none --clip-overlap'), was used to align reads to the human reference genome (NCBI Build 38). Reads aligning with substantial sequence homology to the MT chromosome or to the ENCODE blacklisted regions were omitted from downstream analyses. The ENCODE pipeline standards were used to quantify chromatin accessibility from paired reads derived from non-duplicate sequencing fragments with minor modifications as follows. Macs2[57] was used to call peaks to identify accessible genomic locations using insertion-centered pseudo-fragments (73 bp - community standard) generated on the basis of the start positions of the mapped reads. Briefly, peaks were called on a group-level pooled sample containing all pseudo-fragments observed in all samples within each group. Peaks in the pooled sample that were shared among the biological replicates were retained for downstream analysis, using the union of all group-level reproducible peaks (https://www.encodeproject.org/atac-seq/#standards). We quantified the chromatin accessibility within each peak for each replicate as the number of pseudo-fragments that overlapped with the peak and used the TMM method[60] to normalize the estimates. Differentially accessible peaks between groups were identified using a linear model implemented with the limma R package (version 3.38.3)[61] and incorporating precision weights calculated with the voom function in the limma R package[62]. Chromatin accessibility peaks were considered significantly different across groups if we observed an absolute log₂ fold-change > 1 (estimated from the model coefficients) associated with an FDR adjusted P value < 0.05. HOMER[63] (version 4.7) was used to identify enriched motifs in these peaks. The Integrative Genomics Viewer (IGV) was used to visualize tracks.

**RNA-seq**. NCI-H1944 cells expressing LACZ, SMARCA4 WT or K785R mutant were transduced with nontargeting control or *SMARCA2*-targeting shRNAs in pLKO-based vector (see Lentiviral constructs and infection for sequences). 48 h post transduction, cells were selected with puromycin for 72 h. Cells were scraped and total RNA was extracted using RNeasy Plus Mini Kit (Qiagen) and treated with RNase-free DNase (Qiagen). 3 replicate samples were collected for each treatment condition. The concentration of RNA was determined using NanoDrop 8000 (Thermo Scientific). Approximately 500 ng of total RNA was used as an input for library preparation using TruSeq RNA Sample Preparation Kit v2 (Illumina). The libraries were multiplexed and sequenced on the Illumina HiSeq4000 (Illumina). An average of 52 million single-end 50 bp reads were obtained per sample.

Reads were first aligned to ribosomal RNA sequences to remove ribosomal reads. The remaining reads were aligned to the human reference genome (NCBI Build 38) using GSNAP[56] version '2013-10-10', allowing a maximum of two mismatches per 50 base pair sequence (parameters: '-M 2 -n 10 -B 2 -i 1 -N 1 -w 200000 -E 1 --pairmax-rna=200000 --clip-overlap'). Transcript annotation was based on the Ensembl based GENCODE gene models (GENCODE 27). To quantify gene expression, the number of reads mapped to the exons of each RefSeq gene was calculated using the HTSeqGenie R package. Read counts were scaled by

library size, quantile normalized and precision weights calculated using the "voom" R package[62]. Subsequently, differential expression analysis on the normalized count data was performed using the "limma" R package[61] by contrasting SMARCA4 mutant samples with control samples, respectively. Gene expression was considered significantly different across groups if we observed an |log$_2$ fold-change| $\geq$ 1 (estimated from the model coefficients) associated with an FDR adjusted $P$ value $\leq$ 0.05. In addition, gene expression was obtained in form of normalized Reads Per Kilobase gene model per Million total reads (nRPKM) as described previously[64].

**Beta analysis**. We associated accessible chromatin regions with nearby genes using BETA (version 1.0.7)[65]. The BETA minus mode was used to calculate the regulatory potential (determined through a distance-weighted measure) of specific sets of peaks within a certain distance to a target gene. The BETA basic mode allowed us to integrate differential expression with chromatin openness to evaluate whether the direct effect of changes in the chromatin landscape is promoting or repressing gene expression. In this mode all genes within 100 kb of a peak set are ranked (and listed along the $x$-axis) based on the regulatory potential using the ATAC-seq data. Subsequently, expression information is used to divide genes into SMARCA4 mutant down-regulated (purple line), SMARCA4 mutant up-regulated (red line) and transcriptionally unchanged (dashed line) genes. A one-tailed Kolmogorov-Smirnov test[66] was used to determine whether the up-regulated and down-regulated groups differed significantly from the group of transcriptionally unchanged genes.

**Statistics and reproducibility**. Prism 8 (version 8.3.0) and R (version 3.5.1) were used to generate graphs and run statistical analyses. See individual Methods sections for specific statistical methods. FRET and gel shift assays were replicated twice, with each orthogonal method confirming the same result. SMARCA4 immunoprecipitations followed by silver stains and immunoblots were replicated at least twice. qPCR of gene induction after SMARCA4 WT- and mutant-reconstitution was replicated at least twice and confirmed in 2 different cell lines. qChIP experiments were replicated at least 3 times with similar results. Incucyte confluence measurements and colony forming assays in SMARCA4 WT- and mutant-reconstituted cell lines with and without *SMARCA2* knockdown were replicated at least 3 times. Colony forming assays and immunofluorescence in SMARCA4 WT- and mutant-reconstituted cells after CRISPR knockout of *SMARCA2* were replicated twice. ATAC- and ChIP-seq were performed in duplicate; RNA-seq were performed in triplicate. ChIP- and RNA-seq was validated in a panel of genes using qChIP and qPCR experiments. ATAC-seq and RNA-seq after SMARCA4 WT reconstitution was performed in 2 different cell lines, showing similar results.

**Reporting summary**. Further information on research design is available in the Nature Research Reporting Summary linked to this article.

## Data availability

All ATAC/ChIP/RNA-seq data that support the findings of this study have been deposited in the Gene Expression Omnibus (GEO) with accession code "GSE144844 [https://www.ncbi.nlm.nih.gov/geo/query/acc.cgi?acc=GSE144844]". Full variant information for ~18,000 samples have been deposited in the Genomics Data Commons (GDC) with study accession "phs001179 [https://gdc.cancer.gov/about-gdc/contributed-genomic-data-cancer-research/foundation-medicine/foundation-medicine]". In an effort to minimize any risk of re-identification of individuals with respect to the Health Insurance Portability and Accountability Act, additional detailed data will not be provided. However, all *SMARCA4* variants described in this study are found in Supplementary Data 1. Source data for Fig. 4a, f, g and Supplementary Figs. 5b and 6h are provided with this paper. The remaining data are available within the Article, Supplementary Information or available from the authors upon request. Source data are provided with this paper.

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

## Acknowledgements

We thank the members of the Yauch lab, Andrea Cochran, Tatjana Petojevic and Mariano Oppikofer for technical assistance and thoughtful discussions throughout the course of this study. We thank Ben Haley, Keith Anderson and Jean-Philippe Fortin hub for providing shRNA and guide RNA sequences. We thank Lisa Belmont and Kristi Elkins for contributing supplemental data. We thank the Genentech NGS group for generating RNA-seq libraries and sequencing.

## Author contributions

R.L.Y. conceptualized the project. T.M.F. and R.L.Y. designed experiments, analyzed data and wrote the manuscript. T.M.F. generated figures and performed experiments. R.P. and R.B. performed bioinformatics analyses. E.S.S., S.E.T. and S.M. curated the Foundation Medicine data. Q.Z. and H.T. performed survival analyses. M.K. generated the homology model. S.C. and Z.M. generated ATAC-seq libraries and performed sequencing runs. T.J. performed long term growth assays. All authors reviewed and edited the manuscript.

## Competing interests

T.M.F., R.P., R.B., Q.Z., H.T., S.M., M.K., S.C., Z.M., T.J. and R.L.Y. are employees of Genentech and own shares of Roche. E.S.S. and S.E.T. are employees of Foundation Medicine and own shares of Roche.
