## [Peer Review File · Nature Communications]

REVIEWER COMMENTS

Reviewer #1 (Remarks to the Author): Expert in SWI/SNF epigenomics

Summary:

Previous works have identified SMARCA2 as synthetic lethal in SMARCA4 mutant cancers. This has led to efforts to develop SMARCA2 inhibitors. Given the wide variety of SMARCA4 mutations present in cancer, both heterozygous and homozygous, Fernando et al. set out to define the functional consequences of these mutations, determine how they affect SMARCA2 function, and ultimately to identify the mutational profile of cancers most likely to benefit from therapeutic targeting of SMARCA2. The authors utilized exome sequencing from a 131,668-patient cohort to investigate different types of SMARCA4 mutations found in cancer. One of the most important findings of this paper are the new hotspots in SMARCA4 for missense mutations in cancers not previously identified by The Cancer Genome Atlas. They found that these SMARCA4 mutations are mutually exclusive to targetable drivers such as EGFR, ALK, ROS1 and BRAF. Additionally, they found that NSCLC patients whose tumors have SMARCA4 mutations had a significantly worse response to checkpoint inhibitors. Together these findings identify SMARCA4 mutant cancers as lacking in other therapeutic options and thus a high priority. Mutations were recurrently found in the ATP binding pocket, the DNA binding pocket, and the brace helices of SMARCA4, which suggest a defect in remodeling activity is driving the cancer. Indeed, they report that expressing mutant SMARCA4s reduced remodeling activity and chromatin accessibility when compared to wildtype, but they did not affect assembly/composition of the SWI/SNF complex. Investigating mechanism, in vitro remodeling assays revealed all SMARCA4 mutants to be entirely defective although ATAC-seq showed that some of the missense mutations can partially enhance accessibility. Those same mutants that can partially enhance accessibility are ones that can partially rescue growth after SMARCA2 knockdown. By integrating ChIP-seq, the authors conclude that homozygous loss of SMARCA4 results in SMARCA2 binding at those sites and consequently a synthetic lethal dependency on SMARCA2. In contrast, the point mutant SMARCA4s are dominant negative and exclude SMARCA2 binding thus explaining the reduced/absent dependence upon SMARCA2 in these cancers.

Collectively, the experiments and analyses in the manuscript define relationships between SMARCA4 and SMARCA2; define the biochemical and functional effects of a variety of cancer-associated SMARCA4 mutations, and specifically identify the mutational profile of the cancers most likely to benefit from targeted inhibition of SMARCA2. Overall, I find the experiments well done, the writing clear, and the proposed model mechanistically insightful, and the conclusions provide guidance for the development and deployment of SMARCA2 inhibitors.

Major Points:

None.

Minor Points:

The authors should comment upon the specificity of the SMARCA2 antibody: does it only recognize

SMARCA2, or might it recognize SMARCA4 as well? The authors have only performed chip-seq for SMARCA2 when SMARCA4 is absent. It would be interesting, but not essential, to know where SMARCA2 binds when WT and mutant SMARCA4 is expressed. They interpret Figure 4E to support a model where mutSMARCA4s impair the ability of SMARCA2 to open certain regions of chromatin (dominant negative). The figure shows sites where accessibility is lost (cluster 1) and unchanged (cluster 2, 3, 4) when mutSMARCA4 is re-expressed. When there's no SMARCA4, SMARCA2 is bound at all these regions. When WT SMARCA4 is re-expressed, it binds all of these regions too. They suggest that mutSMARCA4 is displacing SMARCA2 and then is unable to open chromatin. It would be nice to have ChIP-seq for SMARCA2 to show whether it is displaced or rather co-bound. However, I have listed this as a minor comment because I don't think the answer will affect the integrity of what they've already shown – perhaps more of refinement of model presentation and/or interesting next step. I do not view gathering such data as essential.

I found the sentence that begins on line 218 (“Surprisingly, the loss in accessibility...”) quite difficult to follow. I needed to read it several times. It should be clarified.

Supplemental 3C: would benefit from a negative control non-SWI/SNF protein as all lanes are positive in all cases. Nonetheless, the data is believable.

Reviewer #2 (Remarks to the Author): Expert in SWI/SNF

Fernando, et al.

Therapeutic potential of SMARCA4 variants revealed by targeted exome-sequencing of 131,668 cancer patients.

The authors have examined the popular hypothesis that the Brm ATPase is a therapeutic vulnerability for Brg ATPase homozygous mutant tumors and vice versa. They find that while many of the Brg ATPase mutants lacking remodeling activity are not able to compensate for a loss of Brm, some of the mutants are able to compensate even though they have almost no detectable remodeling activity. For example, in figure 4 and 5 see A1186T, 1159 and R973L, which have no detectable remodeling activity yet still compensate for a lack of Brm. This indicates that some additional variable is needed to treat these patients if a good selective Brm inhibitor were available.

The studies are clearly described and well performed and I see no significant issues with Figures 1-5.

The studies do evoke a question of why the remodeling dead mutants compensate. Redundancy with another ATP-dependent chromatin remodeler seems very unlikely in that they are using three replicates in the same cell line in Figure 5 to examine all of the mutants. Another possibility is that remodeling activity is not really the correct activity to identify, and previous studies have indicated that in multicellular organisms, PRC1 opposition is the biologically meaningful activity (PMID:1346755) and seems more related to the pathogenetic mechanisms of SWI/SNF complexes in cancer and neurodevelopmental disease PMID:27941796 and PMID:23540691. Interestingly the remodeling dead G1159V mutation is one of the most common hot spots in the ATPase domain,

indicating that its mechanism of action might be critical. It would be of interest to know if this mutation could rapidly evict polycomb repressive complex 1.

In summary, the authors have provided a clear advance in the understanding of personalized treatment of a group of common cancers having homozygous Brg mutations. If I felt they could get into the lab and address the issue of the mechanism, I would ask for a bit of dessert at the end of the manuscript, but feel that for now the advance is sufficient to help direct the efforts of groups making Brm inhibitors.

Reviewer #3 (Remarks to the Author): Expert in SWI/SNF

Fernando et al, report the mutational profile of SMARCA4 in more than 130000 tumors from different origins. They provide information about the type, the context and the clinical relevance of these mutations. In addition, the authors have selected some missense mutations in hotspot and in essential protein domains to test their inactivation nature. In their study they have used different technical approaches and concluded that most of these mutations are inactivating affecting the chromatin remodeling activity of the complex. They also observe that, similar to the truncating or nonsense, most of the missense mutations tested are vulnerable to the depletion of the paralog SMARCA2, underscoring the importance of determining the inactivating nature of non-truncating SMARCA4 mutations. In parallel, the work undertaken by the authors in the analysis of the role of SMARCA2 in SMARCA4-deficient tumors, its genomic occupancy and chromatin accessibility profiles is of interest.

Overall, this is a sound and comprehensive study that will be of interest by researchers working in cancer and in chromatin remodeling or epigenetics.

Main concerns:

-The title is an overstatement and can be misleading since, to this date, there are not therapeutics available for SMARCA4-tumors.

-For most functional/genomic analysis the author tested the H1944 lung cancer cell line and then used other LC cells for validation. This is OK but there is some lack of coherency in the cell lines selected for validation. For example, they start some validations with the H1299 cells (e.g. effects in gene expression upon SMARCA4 restitution in Supp fig 6), but then used different cell lines for other validations. Which is the reason. The authors need to clarify that.

-This paper contains a large amount of experimental data and but it feels like some of the information has not been fully scrutinized. For example, the data on the ATAC-seq following ectopic expression of the different SMARCA4-mutants is only succinctly presented. It will be worth to include each corresponding heatmap in suppl.fig. 5d. Which is the percentage of sites that had gained/loss accessibility following wt or mutant expression as compared to the control? The data on

the ChIP-seq and ATAC-seq for SMARCA4 and SMARCA2 could have been compared with that of other authors. There are several available (e.g. Pan et al Nat Gen 2019), The same with the RNA-seq: which are the functionalities (gene ontologies) of the genes following restitution of SMARCA4 or depletion of SMARCA2?-The lack of function of at least one of the mutations tested here were already assayed by others (e.g. Stanton et al. Nat Gen 2017). How does their results compare to this previous work?

Other comments:

- Some aspects of the data presentation could be improved.
 - In figure 1 c, please indicate the total number of tumors of each type.
 - In supplementary fig 1a, the blue bars contain the information about tumors that are SWI/SNF-wild type. Is that so or is it only for SMARCA4 wt? Since the comparison is with SMARCA4 mutations it would be more appropriate that the blue bars refer only to the SMARCA4-wt tumors.
 - Please indicate the source for the data on the cancer cells included in suppl. fig 1c.
 - The western-blot of sup fig 3c (right-upper panel) are of bad quality, especially those for ARD2, BRD7 and PHF10, these should be repeated.
 - In figure 4b, the information regarding the LACZ control is missing. In this same figure, please indicate the reasons for selecting the genes in figure 4 d, f and g.
 - In the more functional part of the study, the authors use different cancer cell lines. This reviewer recommends identifying them in the figure legends, because sometimes it is difficult to follow which cell line is being used in a given experiment.
- The work is entirely focused in the mutational status of SMARCA4 in cancer, especially in lung cancer. Because of that it would be appropriate to include the references on the first works reporting loss of expression of SMARCA4 (Reisman et al. Oncogene. 2002) and homozygous mutations (Medina et al. Hum Mut. 2008) of SMARCA4 in this type of cancer. In the former paper, the authors report that some lung cancer cells have lost SMARCA4 and SMARCA2. The authors may want to briefly discuss this in the manuscript.

Reviewer #4 (Remarks to the Author): Expert in lung cancer

The manuscript by Fernando et al., "Therapeutic potential of SMARCA4 variants revealed by targeted exome-sequencing of 131,668 cancer patients", describes the landscape of SMARCA4 mutations seen in human cancers, using predominantly data from Foundation Medicine's panel sequencing test. The authors identify both truncating and hotspot mutations. They then go on to study the hotspot mutations, assess their ability to function in nucleosome remodeling, and evaluate their impact on SMARCA2 synthetic lethality.

This is an interesting manuscript that is a significant addition to the literature on SMARCA4 mutation in cancer.

Detailed comments are below (mostly suggestions for revision, but also a couple points of enthusiasm).

Title: “therapeutic potential” is misleading since the variants don’t have any therapeutic potential. “exome sequencing” is not correct as it is mostly panel sequencing. And “cancer patients” is not correct as it is cancers. How about something like “Functional impact of SMARCA4 mis-sense variants revealed by sequencing of 131,668 cancers”, which also brings in the functional studies in the paper?

Cancer of unknown primary: is it mostly NSCLC, given the similarity?

Mutual exclusivity: I don’t think the pan-RTK/Ras/Raf driver analysis makes sense, as the exclusivity of SMARCA4 mutations is predominantly with EGFR and ALK (and probably ROS1 and RET although numbers are small and not statistically significant...). For KRAS and BRAF, the enrichment is small and the ratio is near 1, while for ERBB2 there is some concordance. I think the section should be re-written to focus on the mutual exclusivity where it is both strong and significant. And I wonder if it has more to do with smoking status than other features?

Overall, I am very enthusiastic about this paper, what it tells us about SMARCA4 mutations and function, and what it tells us about targeting SMARCA2. I have always wondered about SMARCA4 point mutations and whether they are really loss of function mutations. After reading this paper, I think that I know the answer!

Point-by-point response to reviewer comments:

Reviewer #1 (Remarks to the Author): Expert in SWI/SNF epigenomics

Summary:

Previous works have identified SMARCA2 as synthetic lethal in SMARCA4 mutant cancers. This has led to efforts to develop SMARCA2 inhibitors. Given the wide variety of SMARCA4 mutations present in cancer, both heterozygous and homozygous, Fernando et al. set out to define the functional consequences of these mutations, determine how they affect SMARCA2 function, and ultimately to identify the mutational profile of cancers most likely to benefit from therapeutic targeting of SMARCA2. The authors utilized exome sequencing from a 131,668-patient cohort to investigate different types of SMARCA4 mutations found in cancer. One of the most important findings of this paper are the new hotspots in SMARCA4 for missense mutations in cancers not previously identified by The Cancer Genome Atlas. They found that these SMARCA4 mutations are mutually exclusive to targetable drivers such as EGFR, ALK, ROS1 and BRAF. Additionally, they found that NSCLC patients whose tumors have SMARCA4 mutations had a significantly worse response to checkpoint inhibitors. Together these findings identify SMARCA4 mutant cancers as lacking in other therapeutic options and thus a high priority. Mutations were recurrently found in the ATP binding pocket, the DNA binding pocket, and the brace helices of SMARCA4, which suggest a defect in remodeling activity is driving the cancer. Indeed, they report that expressing mutant SMARCA4s reduced remodeling activity and chromatin accessibility when compared to wildtype, but they did not affect assembly/composition of the SWI/SNF complex. Investigating mechanism, in vitro remodeling assays revealed all SMARCA4 mutants to be entirely defective although ATAC-seq showed that some of the missense mutations can partially enhance accessibility. Those same mutants that can partially enhance accessibility are ones that can partially rescue growth after SMARCA2 knockdown. By integrating ChIP-seq, the authors conclude that homozygous loss of SMARCA4 results in SMARCA2 binding at those sites and consequently a synthetic lethal dependency on SMARCA2. In contrast, the point mutant SMARCA4s are dominant negative and exclude SMARCA2 binding thus explaining the reduced/absent dependence upon SMARCA2 in these cancers.

Collectively, the experiments and analyses in the manuscript define relationships between SMARCA4 and SMARCA2; define the biochemical and functional effects of a variety of cancer-associated SMARCA4 mutations, and specifically identify the mutational profile of the cancers most likely to benefit from targeted inhibition of SMARCA2. Overall, I find the experiments well done, the writing clear, and the proposed model mechanistically insightful, and the conclusions provide guidance for the development and deployment of SMARCA2 inhibitors.

Major Points:

None.

Minor Points:

The authors should comment upon the specificity of the SMARCA2 antibody: does it only recognize SMARCA2, or might it recognize SMARCA4 as well?

[redacted]

[redacted]

The authors have only performed chip-seq for SMARCA2 when SMARCA4 is absent. It would be interesting, but not essential, to know where SMARCA2 binds when WT and mutant SMARCA4 is expressed. They interpret Figure 4E to support a model where mutSMARCA4s impair the ability of SMARCA2 to open certain regions of chromatin (dominant negative). The figure shows sites where accessibility is lost (cluster 1) and unchanged (cluster 2, 3, 4) when mutSMARCA4 is re-expressed. When there's no SMARCA4, SMARCA2 is bound at all these regions. When WT SMARCA4 is re-expressed, it binds all of these regions too. They suggest that mutSMARCA4 is displacing SMARCA2 and then is unable to open chromatin. It would be nice to have ChIP-seq for SMARCA2 to show whether it is displaced or rather co-bound. However, I have listed this as a minor comment because I don't think the answer will affect the integrity of what they've already shown – perhaps more of refinement of model presentation and/or interesting next step. I do not view gathering such data as essential.

We thank the reviewer for suggesting this experiment. We agree that data demonstrating whether SMARCA2 is displaced or co-bound with SMARCA4 mutants would help us to refine our model as to how exactly SMARCA4 mutants decrease accessibility at SMARCA2-bound sites. Unfortunately, we did not perform these experiments before submitting the paper, and limitations of lab accessibility due to COVID19-related shutdown of Genentech has made these experiments impossible at this time. We thank the reviewer for understanding these hurdles during this trying time and are grateful that this was point was made as a minor comment.

While we can only speculate at this time, recent data has demonstrated that introducing SMARCA2 K755R and R1159Q mutations (equivalent to SMARCA4 K785R and R1189Q, respectively) in ES cells results in decreased accessibility at a subset of sites⁵ (Fig. 3D-E in Gao et al.). This was accompanied with a loss in SMARCA4 ChIP-seq signal when SMARCA2 mutants were expressed despite minimal changes in SMARCA2 enrichment (in fact SMARCA2 enrichment was quite low relative to SMARCA4 enrichment)⁵. We hypothesize that SMARCA4 mutants in our model behave very similarly to the SMARCA2 mutants introduced in ES cells.

I found the sentence that begins on line 218 (“Surprisingly, the loss in accessibility...”) quite difficult to follow. I needed to read it several times. It should be clarified.

Thank you for pointing out this sentence. We certainly do not want to confuse the reader and have re-written the sentence to clarify the point.

Original:

Surprisingly the loss in accessibility observed after *shSMARCA2* in the LOF mutant lines was not as great as in the LACZ control after *shSMARCA2* in both total ATAC read density and the number of sites lost after *shSMARCA2*, suggesting that the LOF mutants partially rescued accessibility of a subset of the SMARCA2 sites (Supplementary Fig. 7f, Fig. 5e).

Revised:

Surprisingly the decrease in accessibility observed after SMARCA2 knockdown was strongest in the LACZ control relative to the LOF mutant lines in both total ATAC read density and the number of sites lost (Supplementary Fig. 7f, Fig. 5e). These results suggest that the LOF mutants partially rescued the accessibility of a subset of SMARCA2-regulated sites.

Supplemental 3C: would benefit from a negative control non-SWI/SNF protein as all lanes are positive in all cases. Nonetheless, the data is believable.

We thank the reviewer for providing this suggestion. While we don't have a negative control, we would like to note that all FLAG immunoprecipitations were performed with LACZ-transduced 293T cells as a control and run alongside with the FLAG-tagged SMARCA4 WT and mutant immunoprecipitations. As expected, the LACZ control lane has no detectable bands in these blots. For the reviewer's reference, we provide the input lanes for independent FLAG immunoprecipitation experiments where we demonstrate the LACZ lanes are positive for all proteins blotted but not present in the FLAG IP (Rebuttal Fig. 2A). Silver staining of FLAG immunoprecipitations also demonstrate the lack of proteins at the expected sizes of BAF complex members in the LACZ control while maintaining the presence of nonspecific bands just below 75 kD and 37 kD (Rebuttal Fig. 2B). Proteins at the expected size of other BAF complex members are not present in the LACZ control lane of the FLAG immunoprecipitations.

Rebuttal Figure 2. LACZ immunoprecipitations serve as controls for SMARCA4 WT and mutant immunoprecipitations. A) Immunoblot of input and FLAG immunoprecipitations in HEK 293T cells transduced with LACZ or SMARCA4 wildtype or mutant constructs. **B)** Silver stain of input and FLAG immunoprecipitations from cells in (A). Nonspecific (NS) bands are present in both LACZ and SMARCA4 wildtype and mutant lanes.

Reviewer #2 (Remarks to the Author): Expert in SWI/SNF

Fernando, et al.

Therapeutic potential of SMARCA4 variants revealed by targeted exome-sequencing of 131,668 cancer patients.

The authors have examined the popular hypothesis that the Brm ATPase is a therapeutic vulnerability for Brg ATPase homozygous mutant tumors and vice versa. They find that while many of the Brg ATPase mutants lacking remodeling activity are not able to compensate for a loss of Brm, some of the mutants are able to compensate even though they have almost no detectable remodeling activity. For example, in figure 4 and 5 see A1186T, 1159 and R973L, which have no detectable remodeling activity yet still compensate for a lack of Brm. This indicates that some additional variable is needed to treat these patients if a good selective Brm inhibitor were available.

The studies are clearly described and well performed and I see no significant issues with Figures 1-5.

The studies do evoke a question of why the remodeling dead mutants compensate. Redundancy with another ATP-dependent chromatin remodeler seems very unlikely in that they are using three replicates in the same cell line in Figure 5 to examine all of the mutants. Another possibility is that remodeling activity is not really the correct activity to identify, and previous studies have indicated that in multicellular organisms, PRC1 opposition is the biologically meaningful activity (PMID:1346755) and seems more related to the pathogenetic mechanisms of SWI/SNF complexes in cancer and neurodevelopmental disease PMID:27941796 and PMID:23540691. Interestingly the remodeling dead G1159V mutation is one of the most common hot spots in the ATPase domain, indicating that its mechanism of action might be critical. It would be of interest to know if this mutation could rapidly evict polycomb repressive complex 1.

In summary, the authors have provided a clear advance in the understanding of personalized treatment of a group of common cancers having homozygous Brg mutations. If I felt they could get into the lab and address the issue of the mechanism, I would ask for a bit of dessert at the end of the manuscript, but feel that for now the advance is sufficient to help direct the efforts of groups making Brm inhibitors.

We agree with the author that it would certainly be interesting to determine if the SMARCA4 mutants profiled in this manuscript can evict PRC1 complexes as previously described by Stanton et al.⁶. Unfortunately, we did not perform these experiments before submitting this manuscript and instead chose to focus on the relationship between SMARCA4 mutants and SMARCA2. As stated by the reviewer, we wanted to characterize the missense SMARCA4 mutations that haven't been previously described to determine their functional impact and provide insight into how these mutants would behave in the context of SMARCA2 inhibition or depletion. It is interesting to note that a publication from the very same group a year later demonstrated that the increased PRC1 occupancy did not occur in the same regions that lost accessibility following expression of SMARCA4 mutants⁷. Instead changes in PRC1 occupancy and accessibility occurred at distinct sites and suggest a more complex and possibly indirect relationship between PRC1 and BAF. Due to the intricate mechanisms that are probably at play, we thought that teasing this apart would be outside the scope of the current manuscript and best suited for an entirely separate manuscript where several figures could be devoted to characterizing this complex interplay.

Reviewer #3 (Remarks to the Author): Expert in SWI/SNF

Fernando et al, report the mutational profile of SMARCA4 in more than 130000 tumors from different origins. They provide information about the type, the context and the clinical relevance of these mutations. In addition, the authors have selected some missense mutations in hotspot and in essential protein domains to test their inactivation nature. In their study they have used different technical approaches and concluded that most of these mutations are inactivating affecting the chromatin remodeling activity of the complex. They also observe that, similar to the truncating or nonsense, most of the missense mutations tested are vulnerable to the depletion of the paralog SMARCA2, underscoring the importance of determining the inactivating nature of non-truncating SMARCA4 mutations. In parallel, the work undertaken by the authors in the analysis of the role of SMARCA2 in SMARCA4-deficient tumors, its genomic occupancy and chromatin accessibility profiles is of interest.

Overall, this is a sound and comprehensive study that will be of interest by researchers working in cancer and in chromatin remodeling or epigenetics.

Main concerns:

-The title is an overstatement and can be misleading since, to this date, there are not therapeutics available for SMARCA4-tumors.

We agree with the reviewer's comments regarding this and have now changed the title accordingly.

Old title:

Therapeutic potential of *SMARCA4* variants revealed by targeted exome-sequencing of 131,668 cancer patients

New title:

Functional characterization of *SMARCA4* variants identified by targeted exome-sequencing of 131,669 tumors

-For most functional/genomic analysis the author tested the H1944 lung cancer cell line and then used other LC cells for validation. This is OK but there is some lack of coherency in the cell lines selected for validation. For example, they start some validations with the H1299 cells (e.g. effects in gene expression upon SMARCA4 restitution in Supp fig 6), but then used different cell lines for other validations. Which is the reason. The authors need to clarify that.

We appreciate the reviewer's attention to the cell lines used. We chose three SMARCA4 mutant cells, NCI-H1944, NCI-H1299 and A549. As the reviewer stated, our workhorse model is the NCI-H1944 cells, where we have produced the majority of the data (cell growth, chromatin accessibility and gene expression). We felt that assessing the reproducibility of the functional effects for specific mutants was most critical and addressed this in the A549 cell line (Supplemental Figure 7 b,c). We also attempted to address this in NCI-H1299 cells as well, however were unable to ectopically express SMARCA4 WT and mutants to levels similar to endogenous SMARCA4-expressing cell lines (as was the case with NCI-H1944 and A549). Although levels of WT and mutant SMARCA4 were comparable in NCI-H1299 cells, they were extremely overexpressed and lead to decreased proliferation with the expression of SMARCA4 WT, a finding which was not replicated in experiments with NCI-H1944, A549 and other models we tested (Rebuttal Fig. 3a-b). Hence, we decided to only utilize the NCI-H1299 cells to assess the transcriptional effects, since we had in-house ATAC-seq and RNA-seq data done in this cell line where we could identify SMARCA4 targets that had changes in accessibility and gene expression.

Rebuttal Figure 3. SMARCA4 WT reconstitution in SMARCA4-mutant lung cancer cell lines. A) SMARCA4 WT and K785R reconstitution in A549, NCI-H1299 and NCI-H1944. SMARCA4 was overexpressed in NCI-H1299 relative to SW1573, a cell line that endogenously expresses WT SMARCA4. B) SMARCA4 WT reconstitution alters cell proliferation in only NCI-H1299 cells and not in other lung cancer cell lines tested including A549, NCI-H1944, NCI-H838, NCI-H1355 and NCI-H1793.

-This paper contains a large amount of experimental data and but it feels like some of the information has not been fully scrutinized. For example, the data on the ATAC-seq following ectopic expression of the different SMARCA4-mutants is only succinctly presented. It will be worth to include each corresponding heatmap in suppl.fig. 5d. Which is the percentage of sites that had gained/loss accessibility following wt or mutant expression as compared to the control?

In Supplementary Fig. 5d, we primarily focused on the accessibility and expression changes after SMARCA4 WT reconstitution. Hence, we focused on the gained accessibility changes observed in SMARCA4 WT-expressing cells in Supplementary Fig. 5d. In lieu of the individual heatmaps the reviewer has suggested, we had already included a consolidated heatmap of all the accessibility changes after SMARCA4 WT and mutant reconstitution in the manuscript's Fig. 4c. Cluster 1 depicts regions that incur decreases in accessibility after the reconstitution of SMARCA4 mutants (with relatively little change after SMARCA4 WT reconstitution). In the bottom half of the heatmap (clusters 2-4), large increases in accessibility can be observed after SMARCA4 WT expression whereas the majority of these regions remain unchanged with SMARCA4 mutant reconstitution. We break this heatmap down to the individual heatmaps asked by the reviewer. Rebuttal Fig. 4 shows these same regions individually for SMARCA4 WT and mutants in heatmaps similar to Supplemental Fig. 5d. Opening regions are those found in clusters 2-4; here only WT increases the accessibility of these regions whereas the mutants largely result in no change. Closing regions are those found in cluster 1; here these regions are largely accessible in the LACZ control and after WT reconstitution. However, SMARCA4 mutant reconstitution results in a closure of these sites. While we include this figure in the rebuttal below, we did not believe that it adds much value to the overall manuscript and would be showing the same data in Fig. 4c twice. However, if the reviewer feels that it's worth including, we can incorporate this within a supplemental figure.

Rebuttal Figure 4. Individual heatmaps of accessibility changes after SMARCA4 WT and mutant reconstitution in NCI-H1944 cells. Heatmap of ATAC-seq at regions with increased accessibility after SMARCA4 WT reconstitution (opening regions) or decreased accessibility after SMARCA4 mutant reconstitution (closing regions). Data are shown for each replicate (R) as normalized peak counts per million genomic DNA fragments (n=2). Rows are rank ordered by ATAC-seq peaks in WT (for opening regions) or LACZ (for closing regions).

The absolute number of sites opened and closed after reconstitution with SMARCA4 WT or mutant expression are shown relative to the LACZ control in a bar plot in the manuscript's Fig. 4b. More than 2500 sites were gained in SMARCA4 WT-expressing cells relative to LACZ-expressing cells; and more than 1500 sites were lost after the expression of SMARCA4 G1232S (the most strongly closing mutant). We modified the legend and y-axis title to be more clear that these are the differentially accessible sites relative to the LACZ control (also a minor point noted below) (see modified Fig. 4b below with accompanying revised legend).

Revised Manuscript Figure 4b. Significantly open and closed sites as measured by ATAC-seq in NCI-H1944 cells expressing SMARCA4 WT or mutants compared to the LACZ control (n=2 per construct). Significance was tested with a moderated t-statistic (two-sided) and *P* values were adjusted for multiple testing with the Benjamini–Hochberg procedure.

The data on the ChIP-seq and ATAC-seq for SMARCA4 and SMARCA2 could have been compared with that of other authors. There are several available (e.g. Pan et al Nat Gen 2019), The same with the RNA-seq: which are the functionalities (gene ontologies) of the genes following restitution of SMARCA4 or depletion of SMARCA2?-The lack of function of at least one of the mutations tested here were already assayed by others (e.g. Stanton et al. Nat Gen 2017). How does their results compare to this previous work?

We thank the reviewer for this suggestion. We have surveyed the literature and found publications that had performed ATAC-, ChIP- and/or RNA-seq after SMARCA4 WT or mutant reconstitution or SMARCA2 knockdown (summarized in Rebuttal Table 1). From this list, we chose to benchmark our data with studies that had replicates and similar experimental design (comparing SMARCA4 reconstitution experiments side-by-side instead of attempting to compare SMARCA4 reconstitution to SMARCA4 knockdown for example). We also chose to exclude any datasets where we would have to compare our human data to mouse data as that can introduce additional layers of complexity. With this in mind, we prioritized our head-to-head analyses with the data in Pan et al. (ATAC/CHIP/RNA-seq with reconstitution of SMARCA4 WT and K785R/T910M mutants in a small cell ovarian carcinoma cell line); Lissanu Deribe et al. (CHIP/RNA-seq with reconstitution of SMARCA4 WT in non-small cell lung cancer cell lines); Safgren et al. (ATAC-seq with SMARCA2 knockdown in a rhabdomyosarcoma cell line); Vangamudi et al. (RNA-seq with SMARCA2 knockdown in non-small cell lung cancer cell lines) (publications in blue in Rebuttal Table 1)^{1,2,4,8}. This allowed for a comparison of our chromatin accessibility, SMARCA4 occupancy and gene expression data to at least two or more previously published datasets for each NGS technique profiled.

In our data, we found that reconstitution of a SMARCA4 mutant cell line (NCI-H1944) with SMARCA4 WT increased accessibility at ~2500 sites (Fig. 4b). However, SMARCA4 mutants closed >1500 sites (Fig. 4b-c), which we found to be maintained open by SMARCA2 (Fig. 4e; Supplementary Fig. 6c). We first asked whether these sites regulated by SMARCA4 WT/mutant reconstitution in our lung cancer cell line were shared with the Pan et al. study where they similarly reconstituted a SMARCA2- and SMARCA4-deficient small cell ovarian carcinoma cancer cell line (BIN-67) with SMARCA4 WT and assayed chromatin accessibility by ATAC-seq⁴. We found that SMARCA4 reconstitution in the BIN-67 line also lead to modest increases in accessibility at sites opened in NCI-H1944 cells transduced with WT (Rebuttal Fig. 5). We found that the SMARCA2-regulated sites closed by SMARCA4 mutants were relatively inaccessible in control (CON)-transduced BIN-67 cells, consistent with this cell line being deficient for both SMARCA2 and SMARCA4. However, these sites gained modest increases in accessibility after SMARCA4 reconstitution in BIN-67 cells, suggesting that SMARCA4 can compensate for SMARCA2 deficiency at these sites (Rebuttal Fig. 5).

	Publication	Model system	Experimental design and comments	NCBI GEO Accession #
ATAC-seq	Pan et al. Nat Gen 2019	Ovarian cell line (BIN-67)	SMARCA4 WT and mut reconstitution	GSE117301
	Safgren et al. J Biol Chem 2020	Rhabdomyosarcoma cell line (RMS13)	SMARCA2 knockdown	GSE143684
	Hodges et al. Nat Struct Mol Biol 2018	Murine ESCs	SMARCA4 WT and mut reconstitution; not in human	GSE98605
	Xue et al. Nat Commun 2019	Lung cell line (NCI-H1703)	SMARCA4 WT reconstitution; no replicates	GSE121755
ChIP-seq	Pan et al. Nat Gen 2019	Ovarian cell line (BIN-67)	SMARCA4 WT and mut reconstitution	GSE117734
	Lissanu Deribe et al. Nat Med 2018	Lung cell line (NCI-H1299)	Inducible SMARCA4 WT reconstitution	GSE109020
	Xue et al. Nat Commun 2019	Lung cell line (NCI-H1703)	SMARCA4 WT reconstitution; no replicates	GSE121755
	Stanton et al. Nat Gen 2017	murine ESCs	PRC1 occupancy after SMARCA4 WT/mut reconstitution	GSE88968
RNA-seq	Pan et al. Nat Gen 2019	Ovarian cell line (BIN-67)	SMARCA4 WT and mut reconstitution	GSE117311
	Lissanu Deribe et al. Nat Med 2018	Lung cell line (NCI-H1299)	Inducible SMARCA4 WT reconstitution	GSE109010
	Vangamudi et al. Cancer Res 2015	Lung cell line (A549)	SMARCA4 WT reconstitution; no replicates	GSE69088
	Vangamudi et al. Cancer Res 2015	Lung cell line (NCI-H1299)	SMARCA2 knockdown	GSE69088
	Stanton et al. Nat Gen 2017	murine ESCs	SMARCA4 KO and WT; not in human	GSE88968
	Hodges et al. Nat Struct Mol Biol 2018	murine ESCs	SMARCA4 WT and G784E reconstitution; not in human	GSE98605

Selected for analysis

Rebuttal Table 1. Summary of publications where SMARCA2 and SMARCA4 were profiled by NGS techniques for chromatin accessibility, chromatin enrichment or gene expression. Publications in blue were selected for comparison with data in Fernando et al.

Rebuttal Figure 5. Heatmap of chromatin accessibility in NCI-H1944 cells transduced with LACZ or SMARCA4 WT in NCI-H1944 cells ($n=2$ per construct, left) and chromatin accessibility in BIN-67 cells transduced with CONTROL, SMARCA4 WT or SMARCA4 mutants from Pan et al⁴. Top panel shows accessibility at sites opened after SMARCA4 WT reconstitution in NCI-H1944 cells. Bottom panel shows accessibility at sites closed after SMARCA4 mutant reconstitution in NCI-H1944 cells. Data are shown as normalized peak counts per million genomic DNA fragments in a 2 kb window around the peak center. Rows are rank ordered by ATAC-seq peaks. R, replicate.

We next asked how the sites found to be modulated after SMARCA4 WT and mutant expression in NCI-H1944 cells were affected after SMARCA4 mutant expression in BIN-67 cells. While SMARCA4 WT reconstitution in BIN-67 cells lead to increased accessibility at WT-gained sites, reconstitution with SMARCA4 mutants K785R and T910M in BIN-67 cells failed to open these sites much like what was observed after reconstitution of SMARCA4 mutants in NCI-H1944 cells (Rebuttal Fig. 5). Accessibility of sites that were found to close after SMARCA4 mutant expression in NCI-H1944 cells were largely unchanged after SMARCA4 mutant expression relative to CON-transduced BIN-67 cells whereas these sites gained modest increases in accessibility after SMARCA4 WT reconstitution. This is likely due to the lack of SMARCA2 expression in BIN-67 cells⁴, as these sites are not maintained accessible basally in this

cell line. For these reasons, we were unable to assess if SMARCA4 mutant expression can close SMARCA2-regulated sites in BIN-67 cells (which are deficient for both SMARCA2 and SMARCA4).

In addition to identifying how accessibility changed after SMARCA4 WT and mutant reconstitution in BIN-67 cells at our sites of interest in NCI-H1944 cells, we also performed an overlap of the differentially accessible sites found in CON, WT and mutant-reconstituted BIN-67 cells and those found opened and closed after WT or mutant expression in NCI-1944 cells. As described in Pan et al., we found that WT expression in BIN-67 cells induced the opening of almost 20,000 new sites not found in CON-transduced BIN-67 cells (Rebuttal Fig. 6a). The majority of accessible regions found in K785R- and T910M-transduced BIN-67 cells (>70%) were also shared with sites found in CON BIN-67 cells while less than 25% of sites were shared with sites exclusively opened after WT reconstitution, suggesting that these mutants are deficient in opening sites. Those sites shared between SMARCA4 WT and mutant but not found in CON are likely the activity-independent sites described by Pan et al. (Rebuttal Fig. 6a)⁴. We found that ~18% of the sites opened in WT-transduced NCI-H1944 cells were overlapped with sites gained in WT-transduced BIN-67 cells albeit not found significant. This overlap was much greater relative to the overlap with sites found in CON-transduced BIN-67 cells (only 12 overlapping sites) (Rebuttal Fig. 6b). Interestingly we also overlapped the regions we found closing after SMARCA4 mutant expression in NCI-H1944 cells. We found a significant overlap with ATAC sites found in CON-transduced and WT-transduced BIN-67 cells (Rebuttal Fig. 6c). While these results suggest that a subset of sites are commonly regulated in both BIN-67 and NCI-H1944 cells, the majority of accessible sites regulated by SMARCA4 in BIN-67 and NCI-H1944 cells are distinct to each model system, likely attributed to tissue-specific roles of SMARCA4 and the fact that BIN-67 cells are deficient in SMARCA2.

Rebuttal Figure 6. Overlap of ATAC-seq peaks. A) Overlap of ATAC-seq peaks found in CONTROL, WT and SMARCA4 mutant-transduced BIN-67 cells in Pan et al. B) Overlap of ATAC-seq peaks found in CON- and WT-transduced BIN-67 cells (Pan et al) with sites that have opened after SMARCA4 WT reconstitution relative to LACZ-transduced NCI-H1944 cells (Fernando et al). C) Overlap of ATAC-seq peaks found in CON- and WT-transduced BIN-67 cells⁴ with sites that have closed after SMARCA4 mutant reconstitution relative to LACZ-transduced NCI-H1944 cells (Fernando et al).

Rebuttal Figure 7. Heatmap of chromatin accessibility in NCI-H1944 cells transduced with LACZ or SMARCA4 WT in NCI-H1944 cells (n=2 per construct, left); SMARCA4 occupancy in BIN-67 cells transduced with SMARCA4 WT from Pan et al.; SMARCA4 occupancy in NCI-H1299 cells transduced with DOX-inducible SMARCA4 expression from Lissanu Deribe et al. Top panel shows accessibility at sites opened after SMARCA4 WT reconstitution in NCI-H1944 cells. Bottom panel shows accessibility at sites closed after SMARCA4 mutant reconstitution in NCI-H1944 cells. Data are shown as normalized peak counts per million genomic DNA fragments in a 2 kb window around the peak center. Rows are rank ordered by ATAC-seq peaks. R, replicate.

Our data suggested that sites opened by SMARCA4 reconstitution was due to SMARCA4 binding assayed by ChIP-seq. We similarly found that the increase in accessibility after SMARCA4 reconstitution in BIN-67 cells (Rebuttal Fig. 5) was accompanied with SMARCA4 occupancy (Rebuttal Fig. 7). Interestingly, we saw higher SMARCA4 enrichment at the highly accessible sites maintained by SMARCA2 (and closed by SMARCA4 mutants) relative to those sites opened after SMARCA4 WT in NCI-H1944 cells. These results suggest that these highly accessible sites maintained by SMARCA2 (and SMARCA4 in BIN-67 cells) could be canonical or high-affinity mSWI/SNF sites that can be regulated by either SMARCA2 and SMARCA4. Furthermore, the WT-gained sites had less SMARCA4 occupancy relative to the SMARCA2-maintained sites, suggesting that these sites may be a subset of lower affinity sites. Lissanu Deribe et al. also profiled SMARCA4 occupancy after inducible SMARCA4 expression in NCI-H1299 cells¹. While the SMARCA4 ChIP-seq performed in Lissanu Deribe et al. did not demonstrate

nearly as robust enrichment over input as Pan et al. and our own studies, modest SMARCA4 enrichment can be observed in DOX-treated cells in the cluster of peaks that are highly accessible basally, whereas less enrichment is observed in the WT-gained sites (Rebuttal Fig. 7). Taken together, these results confirm that sites closed after SMARCA4 mutant expression may be high affinity mSWI/SNF sites that are conserved across multiple tissue types (both ovarian and lung).

Rebuttal Figure 8. Overlap of SMARCA4 ChIP-seq peaks. Overlap of SMARCA4 peaks from NCI-H1944 cells (Fernando et al.) transduced with SMARCA4 WT in the absence (left) or presence of SMARCA2 knockdown (right) with: A) SMARCA4 ChIP-seq peaks in NCI-H1299 cells transduced with SMARCA4¹; B) SMARCA4 ChIP-seq peaks in BIN-67 cells transduced with SMARCA4 WT⁴; C) a consensus peak set that had overlapping peaks of SMARCA4, SS18 and ARID2 (comprising of canonical BAF and PBAF peaks) in BIN-67 cells⁴.

In addition to comparing the accessibility changes we observed after SMARCA4 WT/mutant reconstitution to previously published datasets, we wanted directly compare the genome-wide occupancy of SMARCA4 observed in our WT-transduced NCI-H1944 cells to other SMARCA4 ChIP-seq datasets. For this comparison, we continued to use the Lissanu Deribe and Pan studies. In our dataset, we found ~24,000 SMARCA4 peaks in WT-transduced NCI-H1944 cells (and a similar number was observed in these cells transduced with SMARCA2 shRNAs); Lissanu Deribe et al. observed ~6,000 peaks; and Pan et al. observed ~50,000 SMARCA4 peaks (Rebuttal Fig. 8). We found a significant overlap ($P=0$) of our SMARCA4 ChIP-seq peaks in NCI-H1944 with those found in NCI-H1299 cells from the Lissanu Deribe study with more than 70% of the NCI-H1299 peaks overlapping with peaks found in WT-transduced NCI-H1944 (Rebuttal Fig. 8a, left). This overlap remained significant when looking at the SMARCA4 peaks in WT-transduced NCI-H1944 cells after SMARCA2 knockdown (Rebuttal Fig. 8a, right). While Pan et al.

Rebuttal Figure 9. Accessibility changes after SMARCA2 knockdown in NCI-H1944 cells relative to RMS13 cells. A) Heatmap of SMARCA2 occupancy and accessibility at ATAC-seq sites lost after SMARCA2 knockdown in NCI-H1944 cells (Fernando et al.) transduced with LACZ. Also shown on right is the same regions in RMS13 cells transfected with siINT or siSMARCA2. B) Read density of accessibility in sites from (A). C) Overlap of statistically significant sites lost SMARCA2 knockdown in LACZ-transduced NCI-H1944 and RMS13 cells.

had substantially more SMARCA4 peaks after reconstituting BIN-67 with SMARCA4 WT, we also found a significant overlap ($P=0$) of these peaks with those found in WT-transduced NCI-H1944 cells (58% of NCI-H1944 peaks were shared with BIN-67) (Rebuttal Fig. 8b). Of the ~50,000 SMARCA4 peaks identified in BIN-67 cells, ~32,000 SMARCA4 peaks overlapped with other BAF and PBAF subunit members (SS18 and ARID2, respectively). We found a significant overlap with the SMARCA4 peaks found in WT-transduced NCI-H1944 cells with this consensus BAF/PBAF peak set in BIN-67 cells, with ~50% of NCI-H1944 SMARCA4 peaks being shared (Rebuttal Fig. 8c). These results suggest a high concordance between the regions of SMARCA4 enrichment found after reconstitution in NCI-H1944 cells with other cell lines of differing tumor types.

We found a marked decrease in chromatin accessibility at more than 7,900 sites after SMARCA2 knockdown in NCI-H1944 cells. We wanted to assess if the sites that lost accessibility after SMARCA2 depletion were shared with sites lost after SMARCA2 knockdown in other cell types. For this comparison, we found that SMARCA2 knockdown using siRNA was performed in a rhabdomyosarcoma cell line, RMS13, in Safgren et al.⁸. At the sites lost after SMARCA2 depletion in NCI-H1944 cells, we found that accessibility in RMS13 cells after SMARCA2 siRNAs were unchanged or somewhat decreased in some replicates (Rebuttal Fig. 9a-b). However, density plots demonstrated high variability among the replicates within the Safgren study (Rebuttal Fig. 9b). Because of this, we also looked at the overlap of statistically significant lost ATAC-seq peaks after SMARCA2 knockdown in NCI-H1944 cells and those statistically significant lost after SMARCA2 siRNA in RMS13 cells. We found only 47 peaks that were both lost in NCI-H1944 and RMS13 cells (Rebuttal Fig. 9c). This is not entirely surprising as RMS13 cells are proficient for SMARCA4, and paralog compensation may explain the lack of overlap between our datasets.

Finally, we compared the gene expression changes induced after NCI-H1944 cells were transduced with SMARCA4 WT with 3 other similar studies. These included our own internal gene expression data found in the manuscript performed in NCI-H1944 and NCI-H1299 cells; data from Lissanu Deribe et al. in NCI-H1299 cells where SMARCA4 was inducibly expressed¹; and data from Pan et al. where SMARCA4 WT was expressed in BIN-67 cells⁴ (Rebuttal Fig. 10a-d). We also included here previously published data from our own group where the ovarian adenocarcinoma cancer cell line TOV112D was reconstituted with SMARCA4 WT³ (Rebuttal Fig. 10e) to have a more similar comparison to the Pan et al. dataset also performed in an ovarian cancer line BIN-67 (although of small cell origin). Rather than looking at the individual fold-changes of each gene, we performed gene set enrichment analysis (GSEA) to identify pathways altered using the MSigDB Hallmark collection of gene sets. We found both unique and shared gene sets enriched among the five different studies, but surprisingly we found 5 gene sets enriched with genes upregulated after SMARCA4 reconstitution in 4 or more of the studies (Rebuttal Fig. 10, highlighted in blue) and several more shared across 3 different studies (Rebuttal Fig. 10, highlighted in red). Pathways shared included those involved in epithelial to mesenchymal transition (EMT); KRAS signaling; DNA damage response and inflammatory responses. These were shared across lung and ovarian cancer cell lines, suggesting that SMARCA4 regulates a conserved set of pathways among different tissue types. While few gene sets were shared only among the ovarian cell lines, several gene sets were unique to SMARCA4 reconstitution in lung cancer cell lines. These included pathways involved in metabolism, cell cycle control and apical surface/junction.

We also compared the gene expression changes that occurred after the reconstitution of the ATPase-dead SMARCA4 mutant (K785R) in NCI-H1944 cells and K785R and T910M mutants in BIN-67 cells⁴. In this comparison, we found that genes affected by SMARCA4 mutant expression were more similar between mutants (K785R and T910M) in BIN-67 cells relative to the same mutant in NCI-H1944 cells (K785R) (Rebuttal Fig. 11a-c). MYC targets, G2M checkpoint and DNA repair genes were downregulated after SMARCA4 mutant expression in BIN-67 cells but did not come up as gene sets altered in K785R-expressing NCI-H1944 cells although these genes were downregulated in WT-expressing NCI-H1944 cells. It remains unclear if these differences are due to cell lineage but also could be attributed to SMARCA2 expression (BIN-67 cells do not express SMARCA2 whereas NCI-H1944 cells do).

Rebuttal Figure 10. Gene set enrichment analysis (GSEA) of gene expression changes after SMARCA4 reconstitution. GSEA of gene expression changes after SMARCA4 reconstitution in: A) NCI-H1944 cells (Fernando et al.) relative to the LACZ control; B) NCI-H1299 cells (Fernando et al.) relative to the vector control; C) NCI-H1299 cells relative to the No Dox control¹; D) BIN-67 cells relative to the vector control⁴; E) TOV112D cells relative to the vector control³.

Lastly, we compared the gene expression changes after SMARCA2 knockdown in NCI-H1944 cells and NCI-H1299 cells². In stark contrast to what was seen after SMARCA4 reconstitution, we found gene sets that were only downregulated after SMARCA2 knockdown (Rebuttal Fig. 12), which is expected as both cell lines are SMARCA4-deficient and rely on SMARCA2 to maintain accessibility at sites important for survival. We found many shared pathways downregulated among the NCI-H1944 and NCI-H1299 cells, including inflammatory response, ER early and late genes and EMT (Rebuttal Fig. 12). These gene sets were upregulated after SMARCA4 WT reconstitution (Rebuttal Fig. 10), suggesting these can be regulated by both SMARCA2 and SMARCA4.

In summary, we performed a thorough analysis of how our data compared to previously published work with similar experimental design. We found that SMARCA4 can regulate the accessibility of a shared

Rebuttal Figure 11. Gene set enrichment analysis (GSEA) of gene expression changes after SMARCA4 mutant. GSEA of gene expression changes after SMARCA4 mutant reconstitution in: A) NCI-H1944 cells (Fernando et al.) relative to the LACZ control; B-C) BIN-67 cells relative to the vector control⁴.

Rebuttal Figure 12. Gene set enrichment analysis (GSEA) of gene expression changes after SMARCA2 knockdown. GSEA of gene expression changes after SMARCA2 knockdown in: A) NCI-H1944 cells (Fernando et al.) relative to the shNTC control; B) NCI-H1299 cells relative to the shLUC control². Gene sets in purple are shared among the two datasets.

subset of genes among different tumor types (at least in lung and ovarian cancer lines profiled here). However there still remains a large proportion of sites that are not shared and are tumor-type specific. Such a finding is not unexpected, given the differences in the lineage derivation, as well as SMARCA2 proficiency, across the models utilized in these experiments.

Other comments:

-Some aspects of the data presentation could be improved.

We appreciate the reviewer's constructive criticism and have taken several steps to address his/her concerns. See below where we modified and addressed the reviewer's comments point-by-point.

-In figure 1 c, please indicate the total number of tumors of each type.

We thought adding the total number of tumors would be too difficult to visual in the figure, and other readers may also be interested in the total number of tumor types profiled overall (in Fig. 1a). Therefore, we included a Supplementary Table 2 where we have summarized the total number of tumors profiled in Fig. 1a, the total number of tumors that had SMARCA4 alterations (represented as % prevalence in Fig. 1a), and the total number of tumors with SMARCA4 alterations that could be assessed for zygosity (Fig. 1c).

-In supplementary fig 1a, the blue bars contain the information about tumors that are SWI/SNF-wild type. Is that so or is it only for SMARCA4 wt? Since the comparison is with SMARCA4 mutations it would be more appropriate that the blue bars refer only to the SMARCA4-wt tumors.

We thank the reviewer for this suggestion and now have performed this analysis comparing tumor mutation burden of SMARCA4-altered patients to SMARCA4 WT patients and replaced this graph.

We have made a point in the first submission of the manuscript that tumor mutation burden (TMB) in NSCLC was not significantly different between patients with 1 SMARCA4 alteration and mSWI/SNF wildtype. With our new analysis, we find that indeed patients with SMARCA4 alterations have higher TMB than SMARCA4 WT patients. We have now modified the manuscript accordingly (modified sentences are highlighted on pg 2, pg 3-4), and we do not believe this alters the main message of the paper.

Revised Manuscript Supplementary Figure 1a. Tumor mutation burden of patients with one or multiple alterations in *SMARCA4* relative to *SMARCA4* wildtype patients

-Please indicate the source for the data on the cancer cells included in suppl. fig 1c.

Cancer cell lines were profiled by exome-sequencing by our internal cell line repository, gCell. We have now included the *SMARCA4* variants identified in this analysis in Supplementary Table 3.

-The western-blot of sup fig 3c (right-upper panel) are of bad quality, especially those for ARD2, BRD7 and PHF10, these should be repeated.

Unfortunately, the current COVID19 situation has made it difficult for us to redo these immunoblots at this time. Overall, given that the results presented in the supplemental figure have little influence on the overall message of the paper, we hope that the reviewer can take this into consideration.

-In figure 4b, the information regarding the LACZ control is missing. In this same figure, please indicate the reasons for selecting the genes in figure 4 d, f and g.

We apologize for not being clear in the manuscript. The data in Fig. 4b are the differentially accessible sites (both lost and gained) relative to the LACZ control. For example, ~2500 sites opened in NCI-H1944 cells when comparing the accessibility of SMARCA4 WT cells relative to cells expressing the LACZ control. For these reasons, the LACZ control was not itself plotted on this graph. We have additionally clarified this in the Fig. 4b legend (see above for the revised figure and legend). In addition, the LACZ data is included above in Rebuttal Fig. 4.

In Fig 4d, f, g, we wanted to show that the SMARCA4 mutants were defective in their ability to induce the transcriptional changes observed in cells expressing SMARCA4 WT. For these reasons, we primarily chose genes that had gained accessibility in SMARCA4 WT-expressing cells relative to the LACZ control in our ATAC-seq experiments; had robust SMARCA4 enrichment at these changed ATAC sites as measured in our ChIP-seq studies, suggesting that they were direct SMARCA4 targets; and finally demonstrated an increase in expression in SMARCA4 WT-expressing cells relative to the LACZ control in our RNA-seq experiments. The genes shown in Fig. 4d, f, g were all genes that fit these criteria. For Fig. 4d, g, we wanted to assess if the SMARCA4 mutants were able to bind chromatin as well as SMARCA4 WT. To this end, we chose the top genes of this panel with the most robust SMARCA4 enrichment in our ChIP-seq studies to assay all the mutants via qChIP to reduce technical challenges frequently associated with this technique.

-In the more functional part of the study, the authors use different cancer cell lines. This reviewer recommends identifying them in the figure legends, because sometimes it is difficult to follow which cell line is being used in a given experiment.

We thank the reviewer for this suggestion. We have gone through and made sure the cell lines used in every panel are clearly identified in the figure legend in both the main manuscript and supplemental figures.

-The work is entirely focused in the mutational status of SMARCA4 in cancer, especially in lung cancer. Because of that it would be appropriate to include the references on the first works reporting loss of expression of SMARCA4 (Reisman et al. *Oncogene*. 2002) and homozygous mutations (Medina et al. *Hum Mut.* 2008) of SMARCA4 in this type of cancer. In the former paper, the authors report that some lung cancer cells have lost SMARCA4 and SMARCA2. The authors may want to briefly discuss this in the manuscript.

We have now included the references pointed out by the reviewer.

Original sentence:

Genomic studies performed in cancer patients have identified a high frequency of alterations in components of the mammalian switch/sucrose non-fermentable (mSWI/SNF or BAF) chromatin remodeling complex, including its core catalytic subunit, SMARCA4¹⁻⁴.

Revised sentence:

Genomic studies performed in cancer patients and tumor-derived cell lines have identified a high frequency of alterations in components of the mammalian switch/sucrose non-fermentable

(mSWI/SNF or BAF) chromatin remodeling complex, including its core catalytic subunit, SMARCA4^{6,9-13}. (Citations 5 and 6 are Reisman et al. and Medina et al., respectively).

The reviewer brings up an important point regarding the potential concurrent loss of SMARCA2 in SMARCA4-deficient cancers that we did not adequately address. Unfortunately, we do not have expression data associated with the mutational data presented in this manuscript and hence cannot directly address this point. Published immunohistochemistry studies evaluating SMARCA4 and SMARCA2 protein expression have variably described the concurrent loss of these proteins, ranging from approximately 1-10% of NSCLCs^{14,15}. In ongoing immunohistochemistry studies from our own group using antibodies validated for their specificity, we observe the concurrent loss of SMARCA2 in ~13% of SMARCA4-deficient (by IHC) lung tumors (n=23). Interestingly, the concurrent loss of SMARCA2 and SMARCA4 has been described in lung tumors of an undifferentiated rhabdoid morphology referred to as SMARCA4-deficient thoracic sarcomas^{16,17} and/or SMARCA4-deficient lung sarcomatoid tumors¹⁸. Disease ontology data associated with the Foundation Medicine data reported in this manuscript indicated <1% (n=234) of the 29,110 lung tumors profiled represented sarcomatoid carcinomas, hence representing a very minor fraction of the specimens assessed (Supplemental Figure 1d). Thoracic sarcomas were not specifically defined but are extremely rare. Although we cannot definitively determine the % of SMARCA4-mutant lung carcinoma specimens that may be deficient in SMARCA2, the current data would suggest that it likely represents a small subset. Nevertheless, it is important to point out the limitations of our assessment, and we have subsequently added commentary to the discussion of the paper to address this point.

New sentence:

Finally, one limitation of this study is our ability to address the potential for concurrent loss of SMARCA2 in SMARCA4-mutant cancers. The previously described association of SMARCA2 loss with rare BAF-deficient sarcomas^{28,29} and/or lung sarcomatoid carcinomas²⁷ (the latter of which represented <1% of the lung cancers profiled, Supplementary Fig 1d) would suggest it represents a minor percentage of SMARCA4-mutant cases^{30,31}, but nevertheless testing for SMARCA2 expression should be considered for future SMARCA2-targeted therapies.

Reviewer #4 (Remarks to the Author): Expert in lung cancer

The manuscript by Fernando et al., “Therapeutic potential of SMARCA4 variants revealed by targeted exome-sequencing of 131,668 cancer patients”, describes the landscape of SMARCA4 mutations seen in human cancers, using predominantly data from Foundation Medicine’s panel sequencing test. The authors identify both truncating and hotspot mutations. They then go on to study the hotspot mutations, assess their ability to function in nucleosome remodeling, and evaluate their impact on SMARCA2 synthetic lethality.

This is an interesting manuscript that is a significant addition to the literature on SMARCA4 mutation in cancer.

Detailed comments are below (mostly suggestions for revision, but also a couple points of enthusiasm).

Title: “therapeutic potential” is misleading since the variants don’t have any therapeutic potential. “exome sequencing” is not correct as it is mostly panel sequencing. And “cancer patients” is not correct as it is cancers. How about something like “Functional impact of SMARCA4 mis-sense variants revealed by sequencing of 131,668 cancers”, which also brings in the functional studies in the paper?

We thank the reviewer for this suggestion. We have changed the title to point out the functional characterization of the mutants. We believe we are being accurate stating that they were identified by targeted exome-sequencing as this was performed with a panel of 394 genes.

Old title:

Therapeutic potential of *SMARCA4* variants revealed by targeted exome-sequencing of 131,668 cancer patients

New title:

Functional characterization of *SMARCA4* variants identified by targeted exome-sequencing of 131,669 tumors

Cancer of unknown primary: is it mostly NSCLC, given the similarity?

While we do not have corroborative testing of cancers of unknown primary (CUPs) in this cohort to infer the tissue of origin (ex: Cancer Type ID[®]), we looked at the enrichment of a mutational signature previously known to associate with smoking status (Zehir et al., *Nat Med* 2017) and the prevalence of canonical lung cancer drivers to determine if they enrich for lung cancer features. Compared to other tumor types that had a low prevalence of the tobacco mutational signature (0.6%), CUPs and NSCLC were 7.3% and 19.1% positive for the tobacco signature. Interestingly CUPs were only 15.7% positive for known/likely alterations in lung cancer drivers (*BRAF*, *EGFR*, *MET*, *ERBB2*, *ALK*, *RET*, *ROS1*) similar to the prevalence seen in other tumor types (17.1%). These two groups were distinct from NSCLC where there was a much higher frequency of alterations in the canonical lung drivers (35.3%). These results suggest that there could be an enrichment of NSCLC in our CUP cohort but they likely include other tumor types and the lung cancers that are present in CUPs may be driven by other oncogenes.

Mutual exclusivity: I don’t think the pan-RTK/Ras/Raf driver analysis makes sense, as the exclusivity of SMARCA4 mutations is predominantly with EGFR and ALK (and probably ROS1 and RET although numbers are small and not statistically significant...). For KRAS and BRAF, the enrichment is small and the ratio is near 1, while for ERBB2 there is some concordance. I think the section should be re-written to focus on the mutual exclusivity where it is both strong and significant. And I wonder if it has more to do with smoking status than other features?

We thank the reviewer for this suggestion. We would like to point out that we did not make any strong points about *KRAS* and *BRAF* being mutually exclusive with *SMARCA4* alterations. We included them in the list of canonical targetable oncogenes that we did find were mutually exclusive with *SMARCA4* mutations when looking at this group of alterations as whole. However, we do concede this mutual

exclusivity is likely driven by the other oncogenes on this list like *EGFR*, *ALK*, *ROS1* and *RET* that were both strongly and significantly mutually exclusive with *SMARCA4* mutations (having an $OR < 0.55$, $P < 0.05$). This is evident when we found that the *KRAS* prevalence in *SMARCA4* mutant tumors (25.09%) was very similar but lower than the overall prevalence found in NSCLC (27.95%). The difference of 2.8% is subtle and drives the marginally significant OR difference.

We have changed this sentence as the reviewer suggests to remove *KRAS* and *BRAF* so as not to mislead the reader.

Original:

Surprisingly *SMARCA4* mutations were mutually exclusive with the most prevalent, targeted oncogenes in NSCLC, including *KRAS*, *EGFR*, *ALK*, *MET*, *BRAF*, *ROS1* and *RET* ($P = 1.2E-34$).

Revised:

Surprisingly *SMARCA4* mutations were mutually exclusive with the most prevalent, targeted oncogenes in NSCLC, including *EGFR*, *ALK*, *MET*, *ROS1* and *RET* ($P = 1.2E-34$).

Regarding smoking status, we found that *SMARCA4*-altered NSCLC patients were 33.4% positive for the mutational signature associated with smoking (described in the previous question) similar to the *KRAS*-mutant NSCLC patients, which were 26.0% positive for this tobacco signature. The increased frequency of the tobacco signature was in stark contrast to NSCLC patients that were positive for other oncogenic drivers (*BRAF*, *EGFR*, *MET*, *ERBB2*, *ALK*, *RET*, *ROS1*), which were only 9.7% positive for the tobacco signature as expected since the prevalence of these oncogenes is high in non-smokers. *SMARCA4*-mutant NSCLC patients also had higher prevalence of the tobacco signature compared to the entire NSCLC cohort (19.1%). The difference seen in smoking status do suggest that the mutual exclusivity seen between *SMARCA4*- and *EGFR/ALK/ROS1/RET*-altered patients could be related to differences in lung cancer etiology and smoking status.

Overall, I am very enthusiastic about this paper, what it tells us about *SMARCA4* mutations and function, and what it tells us about targeting *SMARCA2*. I have always wondered about *SMARCA4* point mutations and whether they are really loss of function mutations. After reading this paper, I think that I know the answer!

We thank you the reviewer for his/her enthusiasm!

REFERENCES

1. Lissanu Deribe, Y. *et al.* Mutations in the SWI/SNF complex induce a targetable dependence on oxidative phosphorylation in lung cancer. *Nat Med* **24**, 1047-1057 (2018).
2. Vangamudi, B. *et al.* The SMARCA2/4 ATPase Domain Surpasses the Bromodomain as a Drug Target in SWI/SNF-Mutant Cancers: Insights from cDNA Rescue and PFI-3 Inhibitor Studies. *Cancer Res* **75**, 3865-3878 (2015).
3. Januario, T. *et al.* PRC2-mediated repression of SMARCA2 predicts EZH2 inhibitor activity in SWI/SNF mutant tumors. *Proc Natl Acad Sci U S A* **114**, 12249-12254 (2017).
4. Pan, J. *et al.* The ATPase module of mammalian SWI/SNF family complexes mediates subcomplex identity and catalytic activity-independent genomic targeting. *Nature genetics* **51**, 618-626 (2019).
5. Gao, F. *et al.* Heterozygous Mutations in SMARCA2 Reprogram the Enhancer Landscape by Global Retargeting of SMARCA4. *Mol Cell* **75**, 891-904 e7 (2019).
6. Stanton, B.Z. *et al.* Smarca4 ATPase mutations disrupt direct eviction of PRC1 from chromatin. *Nature genetics* **49**, 282-288 (2017).
7. Hodges, C., Kirkland, J.G. & Crabtree, G.R. The Many Roles of BAF (mSWI/SNF) and PBAF Complexes in Cancer. *Cold Spring Harbor perspectives in medicine* **6**, a026930 (2016).
8. Safgren, S.L. *et al.* The transcription factor GLI1 cooperates with the chromatin remodeler SMARCA2 to regulate chromatin accessibility at distal DNA regulatory elements. *J Biol Chem* **295**, 8725-8735 (2020).
9. Shain, A.H. & Pollack, J.R. The spectrum of SWI/SNF mutations, ubiquitous in human cancers. *PLoS one* **8**, e55119 (2013).
10. Kadoch, C. *et al.* Proteomic and bioinformatic analysis of mammalian SWI/SNF complexes identifies extensive roles in human malignancy. *Nature genetics* **45**, 592-601 (2013).
11. Hodges, H.C. *et al.* Dominant-negative SMARCA4 mutants alter the accessibility landscape of tissue-unrestricted enhancers. *Nature structural & molecular biology* **25**, 61-72 (2018).
12. Reisman, D.N. *et al.* Concomitant down-regulation of BRM and BRG1 in human tumor cell lines: differential effects on RB-mediated growth arrest vs CD44 expression. *Oncogene* **21**, 1196-207 (2002).
13. Medina, P.P. *et al.* Frequent BRG1/SMARCA4-inactivating mutations in human lung cancer cell lines. *Hum Mutat* **29**, 617-22 (2008).
14. Herpel, E. *et al.* SMARCA4 and SMARCA2 deficiency in non-small cell lung cancer: immunohistochemical survey of 316 consecutive specimens. *Ann Diagn Pathol* **26**, 47-51 (2017).
15. Reisman, D.N., Sciarrotta, J., Wang, W., Funkhouser, W.K. & Weissman, B.E. Loss of BRG1/BRM in human lung cancer cell lines and primary lung cancers: correlation with poor prognosis. *Cancer Res* **63**, 560-6 (2003).
16. Le Loarer, F. *et al.* SMARCA4 inactivation defines a group of undifferentiated thoracic malignancies transcriptionally related to BAF-deficient sarcomas. *Nat Genet* **47**, 1200-5 (2015).
17. Sauter, J.L. *et al.* SMARCA4-deficient thoracic sarcoma: a distinctive clinicopathological entity with undifferentiated rhabdoid morphology and aggressive behavior. *Mod Pathol* **30**, 1422-1432 (2017).
18. Rekhtman, N. *et al.* SMARCA4-Deficient Thoracic Sarcomatoid Tumors Represent Primarily Smoking-Related Undifferentiated Carcinomas Rather Than Primary Thoracic Sarcomas. *J Thorac Oncol* **15**, 231-247 (2020).

REVIEWERS' COMMENTS

Reviewer #1 (Remarks to the Author):

The authors have addressed all of my comments to my satisfaction.

Reviewer #3 (Remarks to the Author):

The authors have appropriately answered most of the comments of this reviewer. There was a western-blot that was asked to be repeated but that, due to the current pandemic scenario, the authors could not perform. This is understandable.

Reviewer #4 (Remarks to the Author):

I am satisfied with the revisions and have no further comments.